# Split-BioID a conditional proteomics approach to monitor the composition of spatiotemporally defined protein complexes

Isabel Myriam Schopp[1,2], Cinthia Claudia Amaya Ramirez[1,2], Jerneja Debeljak[1,2], Elisa Kreibich[1,2], Merle Skribbe[1,2], Klemens Wild[2] & Julien Béthune[1,2]

Understanding the function of the thousands of cellular proteins is a central question in molecular cell biology. As proteins are typically part of multiple dynamic and often overlapping macromolecular complexes exerting distinct functions, the identification of protein–protein interactions (PPI) and their assignment to specific complexes is a crucial but challenging task. We present a protein fragments complementation assay integrated with the proximity-dependent biotinylation technique BioID. Activated on the interaction of two proteins, split-BioID is a conditional proteomics approach that allows in a single and simple assay to both experimentally validate binary PPI and to unbiasedly identify additional interacting factors. Applying our method to the miRNA-mediated silencing pathway, we can probe the proteomes of two distinct functional complexes containing the Ago2 protein and uncover the protein GIGYF2 as a regulator of miRNA-mediated translation repression. Hence, we provide a novel tool to study dynamic spatiotemporally defined protein complexes in their native cellular environment.

[1] CellNetworks Junior Research Group Posttranscriptional Regulation of mRNA Expression and Localization, Heidelberg University, INF 328, D-69120 Heidelberg, Germany. [2] Biochemie-Zentrum Heidelberg (BZH), INF 328, D-69120 Heidelberg, Germany. Correspondence and requests for materials should be addressed to J.B. (email: Julien.Bethune@bzh.uni-heidelberg.de).

Physical interactions between proteins are required for most cellular processes, and the identification and validation of protein–protein interactions (PPI) is a usual starting point when characterizing a novel protein. Various methods exist for the identification of potential PPI. A very common approach relies on the affinity purification (AP) of a bait protein from cell lysates, followed by a comprehensive identification of co-purifying proteins by mass spectrometry (MS). However, the analysis of such proteomics data is often complicated by the dynamic nature of protein complexes. Indeed, many proteins belong to multiple functional complexes with distinct or overlapping protein compositions. For instance, Argonaute (Ago) proteins, the central players involved in miRNA-mediated gene silencing[1], are part of at least two functionally distinct complexes. In the miRNA-induced silencing complex (miRISC), Ago is involved in post-transcriptional repression of mRNA function, while in the RISC-loading complex (RLC), Ago gets loaded with miRNAs and interacts with factors stimulating this process. Multiple studies have performed AP-MS approaches using Ago as bait[2–4]. The corresponding data sets include proteins that play an important role at diverse steps of the pathway, such as the RLC components TRBP and Dicer, or the TNRC6 proteins that are core components of the miRISC. The AP-MS approaches, however, suffered from two main limitations: (1) important functional factors such as the CCR4/NOT complex, which is directly recruited by TNRC6 and is required for efficient miRNA-mediated silencing[5–7] were notably absent from the AP-MS data sets, and (2) it is not possible to assign novel identified proteins to a specific step of the pathway as Ago is part of both RLC and miRISC.

To address the former, we decided to use the recently described BioID technique[8]. It is based on a variant of the E. coli biotin ligase BirA. BirA uses adenosine tri-phosphate and biotin to produce reactive biotinyl-5′-AMP that is tightly retained in its active centre, making it only accessible to a specific acceptor peptide. The BirA R118G variant, termed BirA*, has weaker affinity for biotinyl-5′-AMP allowing its release in the cytoplasm[9], leading to the biotinylation of proximate proteins within an estimated 10 nm range[10]. Thus, fusion of BirA* to a bait protein enables biotinylation of vicinal proteins and their isolation on streptavidin-coupled beads. Side-by-side comparison of BioID- and AP-MS revealed that both methods identified relevant proteins but yielded moderately overlapping data sets due to the different bias of both techniques: AP detects rather stable interactions while BioID reflects close proximity within cells. Consequently, BioID proved to be better at detecting weak interactions or proteins with low expression levels[11]. With the different bias of BioID, we reasoned that it could be a viable alternative to identify additional PPI involved in miRNA-mediated silencing. However, since efficient biotinylation occurs over a time scale of 6–24 h (ref. 8), proteins identified by BioID integrate all potential interactions with the bait protein over this period of time. Hence, BioID-like AP-MS does not allow the analysis of specific complexes but rather gives an overview of all possible PPI in which a given protein may be involved.

A protein fragment complementation assay (PCA) is a powerful approach to validate known or putative binary interactions in a native cellular environment. Here a protein with a quantifiable activity is split into two poorly interacting non-functional fragments that can reassemble to restore activity when fused to two interacting proteins. We reasoned that a PCA based on BioID would be a unique assay for PPI that would address limitations of AP/BioID-MS approaches. Indeed, the conditional activation of BirA* on the interaction of two specific proteins allows harnessing the proximity-dependent labelling power of BioID in a much more spatially and temporally defined manner with the potential of selectively biotinylate-specific subcomplexes. We show here that it is indeed possible to split BirA* into two PCA-suitable fragments. The resulting split-BioID assay is a conditional proteomics method that allows validating in a single and simple assay binary PPI, and the concomitant labelling of additional vicinal proteins belonging to the corresponding complex in live cells. Using various examples, we demonstrate that our split-BioID technique is a bona fide PCA. Furthermore, focusing on PPI involved in the miRNA-mediated silencing pathway, we show that it allows very-high-resolution proteomics of functional complexes. Indeed, by selectively activating Ago with either Dicer or TNRC6 using split-BioID, we could specifically analyse the RLC or the miRISC and identify a previously unknown regulator of miRNA-mediated translational repression. Another study recently described an alternative split-BioID assay[12], suggesting that BirA* can be split at multiple sites.

## Results

**Design of split-BioID.** To develop our split-BioID PCA, it was essential to identify two inactive fragments of BirA* that can reassemble into an active enzyme when brought in close proximity (Fig. 1a). To this end, we made use of the FKBP (12-kDa FK506-binding protein) and FRB (FKBP-rapamycin-binding domain) proteins that do not interact in the absence of rapamycin but form a tight ternary complex in its presence[13]. The E. coli BirA enzyme is made of three subdomains (Fig. 1b): an N-terminal domain interacts with DNA and mediates repression of the biotin operon, a central part binds biotin and contains the catalytic site[14], a C-terminal domain interacts with the natural substrate protein and contributes to adenosine tri-phosphate binding[15]. According to published data, we split BirA* at three different sites, and the corresponding N-terminal (NBirA*) and C-terminal (CBirA*) fragments were fused

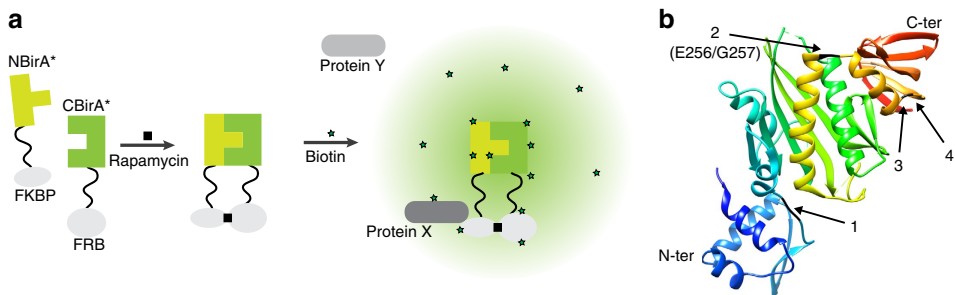

**Figure 1 | Design of split-BioID.** (**a**) Schematic drawing of the proof-of-principle set-up for split-BioID. FRB fused to CBirA* and FKBP fused to NBirA* interact with each other on addition of rapamycin. (**b**) BirA crystal structure (Protein Data Bank 1BIB) showing the four tested splitting sites (1: Q65/L66, 2: E256/G257, 3: N270/F271 and 4: N273/R274), the optimal E256/G257 site is indicated (See Supplementary Fig.1).

to FKBP and FRB, respectively (Fig. 1a,b). Two constructs (N270/F271 and N273/R274) split BirA* in a way that separates the C-terminal domain that was previously described as essential for the activity of BirA[15]. Another construct segregates the N-terminal DNA-binding domain (Q65/L66), which was also reported to stimulate activity[16]. In addition, a fourth construct (E256/G257) was designed guided by structural analysis of the enzyme and splits BirA* before the last helix of its core catalytic domain. The corresponding protein fragments were then co-expressed in HeLa cells, and biotinylation analysed in the absence or presence of rapamycin (Supplementary Fig.1). The best results were obtained for the E256/G257 splitting site that produced two fragments with low activity, as demonstrated by low levels of biotinylation in the absence of rapamycin, that were efficiently reactivated on rapamycin-mediated interaction. This splitting site was thus used for all further experiments. Conveniently, NBirA* and CBirA* can be indifferently appended to the N- and C termini of FRB or FKBP as all combinations of fusion proteins tested (Fig. 2a) led to strong activation of biotinylation on addition of rapamycin (Fig. 2b,c). This suggests that split-BioID can be applied to any proteins provided they can be tagged either on their N- or C termini. Typical for a PCA, the re-assembled fragments showed a reduced relative activity (ca. 2.5%) when compared to BirA* (Fig. 2d). This lower activity was sufficient for performing BioID experiments (see below). A recent study also described a PCA based on two BirA* fragments using an alternative splitting site (E140/Q141)[12]. We wondered if this splitting site would perform similar to our PCA. To test this, we compared both split-BioID assays using our FRB/FKBP system. With this side-by-side comparison, our splitting site appears to perform better as it resulted in a much stronger re-activation on rapamycin-induced dimerization evidenced by the observed stronger biotinylation signal with similar expression levels of the NBirA* and CBirA* fusions (Supplementary Fig. 2).

**Monitoring a phosphorylation-dependent complex.** Split-BioID was next validated on a physiological phosphorylation-dependent PPI that had been previously analysed with a split luciferase PCA[17]. Throughout the cell cycle interphase, the G2/M transition-regulating protein phosphatase Cdc25C is phosphorylated at S216, and subsequently binds 14-3-3ε. This binding sequesters Cdc25C in the cytoplasm, impeding access to its nuclear substrates[18]. We fused CBirA* with either Cdc25C WT or an S216A non-phosphorylatable variant, and fused NBirA* with either the Cdc25C WT-binding partner 14-3-3ε or green fluorescent protein (GFP) as a negative control. Transient co-expression of CBirA*-Cdc25C WT with NBirA*-14-3-3ε yielded a much stronger global biotinylation signal than the Cdc25C S216A/14-3-3ε combination (Fig. 3a,b), similar to the split luciferase PCA that was performed on the same proteins[17]. As expected, the Cdc25C WT/GFP pair produced only background levels of biotinylation. Importantly, immunofluorescence experiments showed that the fusion proteins are all localized in the cytosol (Fig. 4). Of note, we observed that NBirA*-14-3-3ε was much less expressed than endogenous 14-3-3ε, while CBirA*-Cdc25 fusion proteins were overexpressed when compared to endogenous Cdc25C (Supplementary Fig. 3). The same was true when stable cell

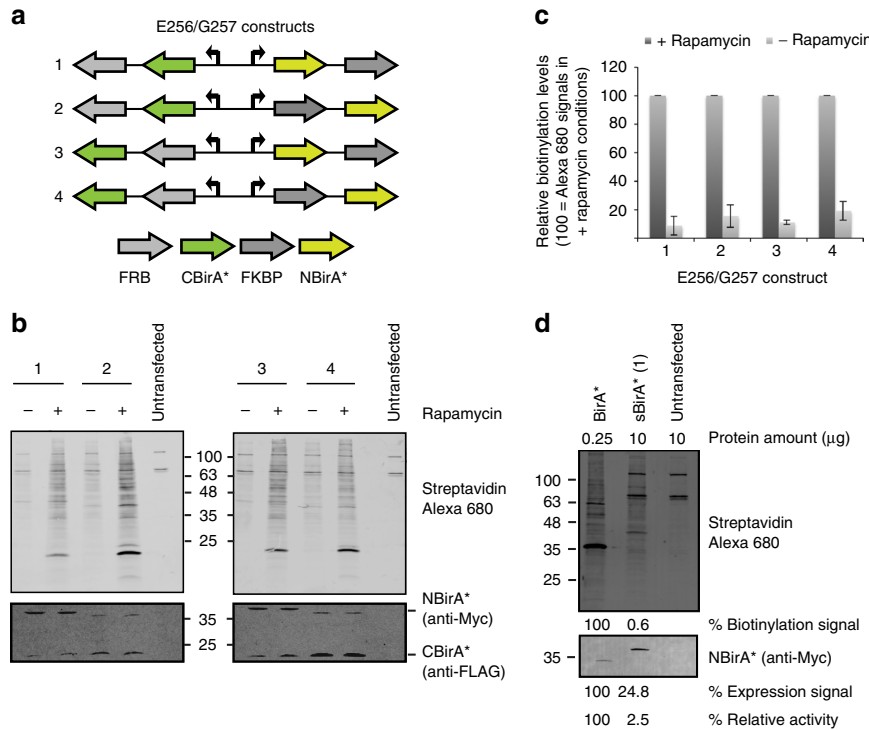

**Figure 2 | Characterization of split-BioID.** (**a**) The four tested constructs, corresponding to all possible orientations of the CBirA*-FRB and NBirA*-FKBP fusions. (**b**) Blots of lysates of HeLa cells transiently transfected with the constructs shown in **a**, and treated with or without rapamycin. Biotinylation was analysed using Alexa680-labelled streptavidin, and expression levels of the fusion proteins with antibodies against FLAG and Myc tags as indicated. (**c**) Quantification of **b**: relative overall biotinylation levels were estimated by integrating Alexa680 signals in each lane in **b** and normalizing them to the signal measured for an endogenous biotinylated protein in the same lane. Signals from non-transfected samples were set to 0% (error bars, s.d.; $n = 3$ independent experiments). (**d**) Activity of the original BirA* compared to the split-BirA* (construct 1). Total biotinylation (streptavidin-Alexa680) and protein expression levels (anti-Myc signal) were set to 100% for BirA*, taking into account the different protein amounts loaded. Relative activity is the ratio of biotinylation over Myc levels.

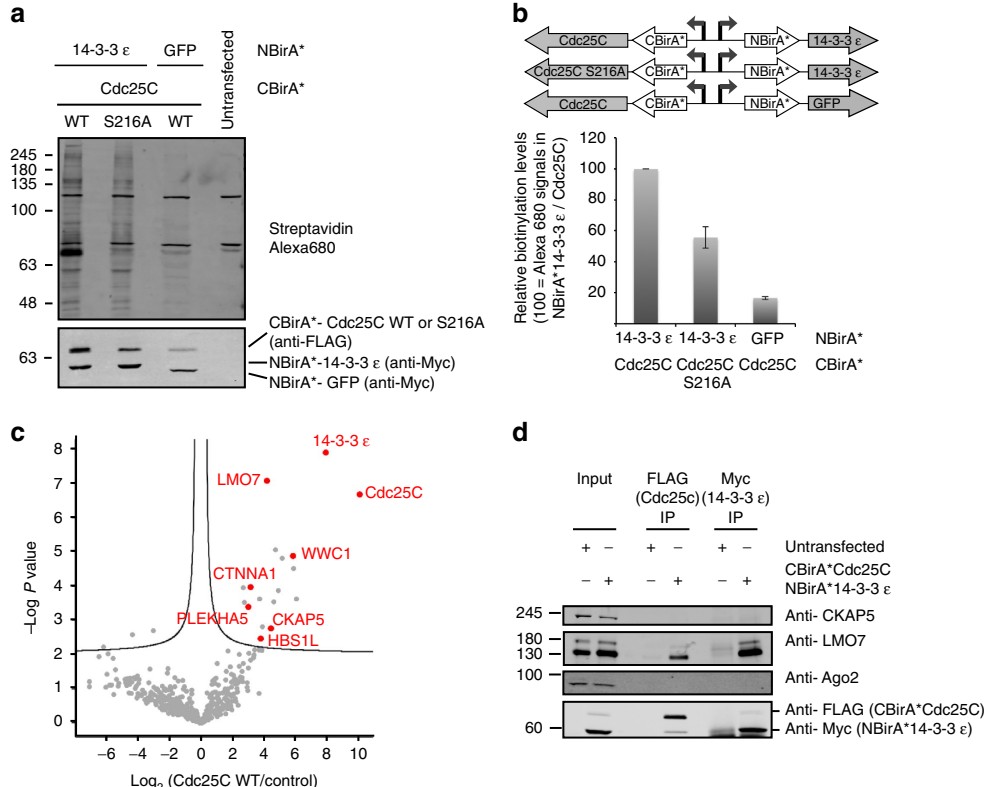

**Figure 3 | Application of split-BioID to a physiological phosphorylation-dependent PPI. (a)** Blots of lysates of cells transiently expressing NBirA*-14-3-3ε/CBirA*-Cdc25c (WT or S216A), or NBirA*-GFP/CBirA*-Cdc25c WT as a control. **(b)** Quantitative analysis of **a** performed as described in Fig. 2c (error bars, s.d.; *n* = 3 independent experiments). **(c)** Volcano plot showing proteins enriched in the Cdc25C WT over the control BioID samples from stable cell lines. The logarithmic ratios of protein LFQs were plotted against negative logarithmic *P* values of a two-sided two samples *t*-test. The hyperbolic curve delimitate significantly enriched proteins from common hits (FDR ≤ 0.07, *n* = 3). Hits that showed higher LFQs than in the GFP and the Cdc25C S216A samples are indicated in red. **(d)** Validation co-IP experiments from HeLa cell lysates transfected with myc-NBirA*-14-3-3ε and FLAG-CBirA*-Cdc25C or lysates of untransfected cells as a control. Endogenous LMO7 is detected in both FLAG and Myc IPs from transfected cells while endogenous CKAP5 is not.

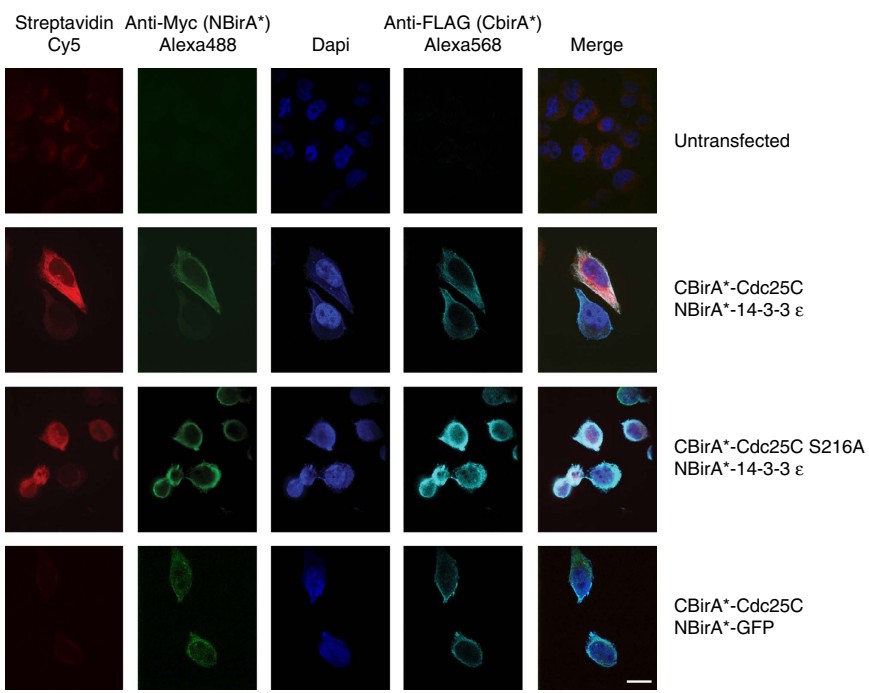

**Figure 4 | Localization of the fusion proteins used for the Cdc25C/14-3-3ε split-BioID.** Immunofluorescence of transiently expressed constructs by Myc- and FLAG-tag detection as well as the detection of biotinylated proteins by Cy5-coupled streptavidin. Scale bar, 15 μm.

lines were constructed using the same plasmids that were used for transient expression, albeit both fusion proteins were expressed at lower levels (Supplementary Fig. 3).

Although, the sequestering role of the 14-3-3ε/Cdc25C interaction may imply that no additional protein should be bound by this pair, we next performed MS analysis and label-free quantification (LFQ) on biological triplicates of the Cdc25C WT/14-3-3ε and Cdc25C S216A/14-3-3ε samples from stable cell lines. Significant enrichment was calculated over a pool of BioID runs on six unrelated proteins (see 'Methods' section) and we further deleted the resulting hits that had LFQ values lower than in the split-BioID GFP control sample. We then look at proteins enriched in the Cdc25C WT/14-3-3ε over the Cdc25C S216A sample. Nine such proteins were found (Fig. 3c and Supplementary Data 1). Consistent with the gain in affinity for 14-3-3 proteins of phosphorylated Cdc25C compared to the non-phosphorylated form (dissociation constant of 57 nM and 1.2 μM (ref. 19), respectively), 14-3-3ε was enriched ca. fourfold and Cdc25C 40-fold in the WT sample. The other seven proteins have no obvious link to the function of Cdc25C but we nevertheless tried to validate two potential interactions (CKAP5 and LMO7) with Cdc25C and 14-3-3ε by co-IP experiments. While we could not find evidence for an interaction with CKAP5, LMO7 was co-immunoprecipitated with (myc-tagged) NBirA*-14-3-3ε and with (FLAG-tagged) CBirA*-Cdc25C (Fig. 3d). Interestingly, during the revision of this paper, a study described an interaction of Lin11, Isl-1 and Mec-3 (LIM) proteins, including LMO7, with Cdc25C and 14-3-3ε. In this study, LIM proteins were shown to both positively regulate phosphorylation of Cdc25C and promote sequestering of Cdc25C by forming a ternary complex with Cdc25C and 14-3-3ε in the cytoplasm[20]. Hence, split-BioID correctly identified LMO7 as interacting with the phosphorylation-dependent Cdc25C/14-3-3ε dimer. It will be interesting to see if some of the other identified proteins are also previously unknown components of the cytoplasmic Cdc25C/14-3-3ε complex. Finally, it is noteworthy that, as predicted, the nuclear targets of Cdc25C, Cdc2 and cyclin B[21] are absent from the MS data. Together, these data demonstrate that split-BioID is a bona fide PCA that can validate binary PPI under different conditions.

**Monitoring maturation of the miRISC**. We next explored the potential of the method for resolving functional subcomplexes involved in miRNA-mediated gene silencing. In the course of the formation of the miRISC, Ago is successively part of two distinct protein subcomplexes containing either Dicer or TNRC6 (Fig. 5a). Dicer processes pre-miRNAs to produce miRNAs that are then loaded onto Ago. When bound to Dicer, Hsp90/Hsc70-stabilized[22] empty Ago is part of an RLC that contains a handful of proteins[23]. The Hsp90/Hsc70 machinery not only stabilizes empty Ago2, but also actively participates in stimulating the transfer of miRNAs to Ago[24]. Once loaded with miRNAs, Ago directly interacts with TNRC6 proteins to form the miRISC that represses translation and stimulates mRNA decay[1], and partially localizes to RNA granules[25,26]. We analysed whether split-BioID could discriminate between these two steps of miRISC assembly. We first tested different fusion proteins of Dicer, and the Ago and TNRC6 paralogs Ago2 and TNRC6C (Fig. 5b and Supplementary Fig. 4a). We found that, when combined to various NBirA* fusion proteins, CBirA* fused to GFP consistently produced significant biotinylation signals that precluded its use as a negative control (Supplementary Fig. 4b). This is in contrast to CBirA* fused to CNOT8 (an indirect interacting partner of Ago2, used as a negative control for the PCA) that yielded background biotinylation signal when combined with NBirA*-Ago2. As a

possible explanation, we noticed that CBirA*-GFP was expressed at much higher levels than any other fusion protein we used in this study. This might drive re-association with the co-expressed NBirA* fusions and lead to increase background. By contrast, a CBirA*-Ago2 fusion led to strong activation when combined to NBirA* fused to either Dicer or TNRC6C, and to background signals when combined to NBirA*-GFP (Fig. 5b). Hence, this combination was identified as the best and was used further. When compared to the endogenous proteins, we observed that CBirA*-Ago2 was less expressed than endogenous Ago2 while NBirA*-Dicer was clearly overexpressed (Supplementary Fig. 4c). The comparison could not be performed for NBirA*-TNRC6C as full-length TNRC6C does not seem to be expressed at significant levels in HeLa cells, which agrees with functional knockdown data performed on the TNRC6 paralogs in HeLa cells[27].

Split-BioID was then performed on transiently transfected cells and the resulting biotinylated proteins were isolated. To test for the general specificity of split-BioID, NBirA*-TNRC6C was immunoprecipitated. We observed that, in addition to CBirA*-Ago2, interaction-induced biotinylated proteins were also found in the precipitated material but not in the control IP (Fig. 5b), strongly suggesting that these proteins are bona fide interaction partners of TNRC6C or Ago2. To determine the resolution of the method, known components of the two different Ago-containing complexes, RLC and miRISC, were probed. TAR RNA-binding protein (TRBP) is a cofactor of Dicer that stimulates miRNA loading onto Ago2 within the RLC[28,29], while the CCR4/NOT complex is recruited to the miRISC through direct interaction of TNRC6 with the CNOT1 subunit[5,6]. Strikingly, TRBP was specifically detected in the Ago2/Dicer split-BioID, while CNOT1 was only detected in the Ago2/TNRC6 split-BioID (Fig. 5c). Both proteins were absent from the Ago2/GFP-negative control sample as well as in the control samples where NBirA*-Dicer and NBirA*-TNRC6C were expressed in the absence of a CBirA* fragment, excluding residual biotinylation activity of both fusion proteins. Hence, in addition to validating binary interactions, split-BioID is also able to specifically identify additional PPI belonging to the corresponding subcomplexes. Finally, IF experiments showed that all fusion proteins were mainly localized to the cytosol and not segregated to different compartments (Fig. 6).

**Probing the RLC and miRISC proteomes**. MS analysis was then performed on biological triplicates of the split-BioID samples, significant enrichment was calculated over a pool of BioID runs on six unrelated proteins (see 'Methods' section) and over the split-BioID GFP control. Altogether, 68 proteins were enriched over control samples (Supplementary Data 2). We then looked at LFQ enrichment in the TNRC6C sample over the Dicer sample (Fig. 7a and Supplementary Data 2). The TNRC6C-enriched proteins (50 proteins, 40% known to be physically or functionally Ago2-associated, including 32% proposed to have miRISC-associated function) excluded RLC-associated components while miRISC-associated factors (Fig. 7b and Supplementary Data 2) were absent in Dicer-enriched proteins (14 proteins, 8 known to be physically or functionally Ago2-associated, including 7 proposed to have a RLC-associated function). As predicted, Dicer is absent from the Ago2/TNRC6C sample while TNRC6C is excluded from the Ago2/Dicer sample. As a comparison we generated a data set of potential interacting proteins by applying classical BioID to Ago2. Similar to AP-MS, BioID-MS (Supplementary Data 3) identified components of both RLC and miRISC including Dicer and TNRC6C. Hence, split-BioID considerably increases the resolution of the assay. Interaction network analysis shows that BioID identified PPI clustering around

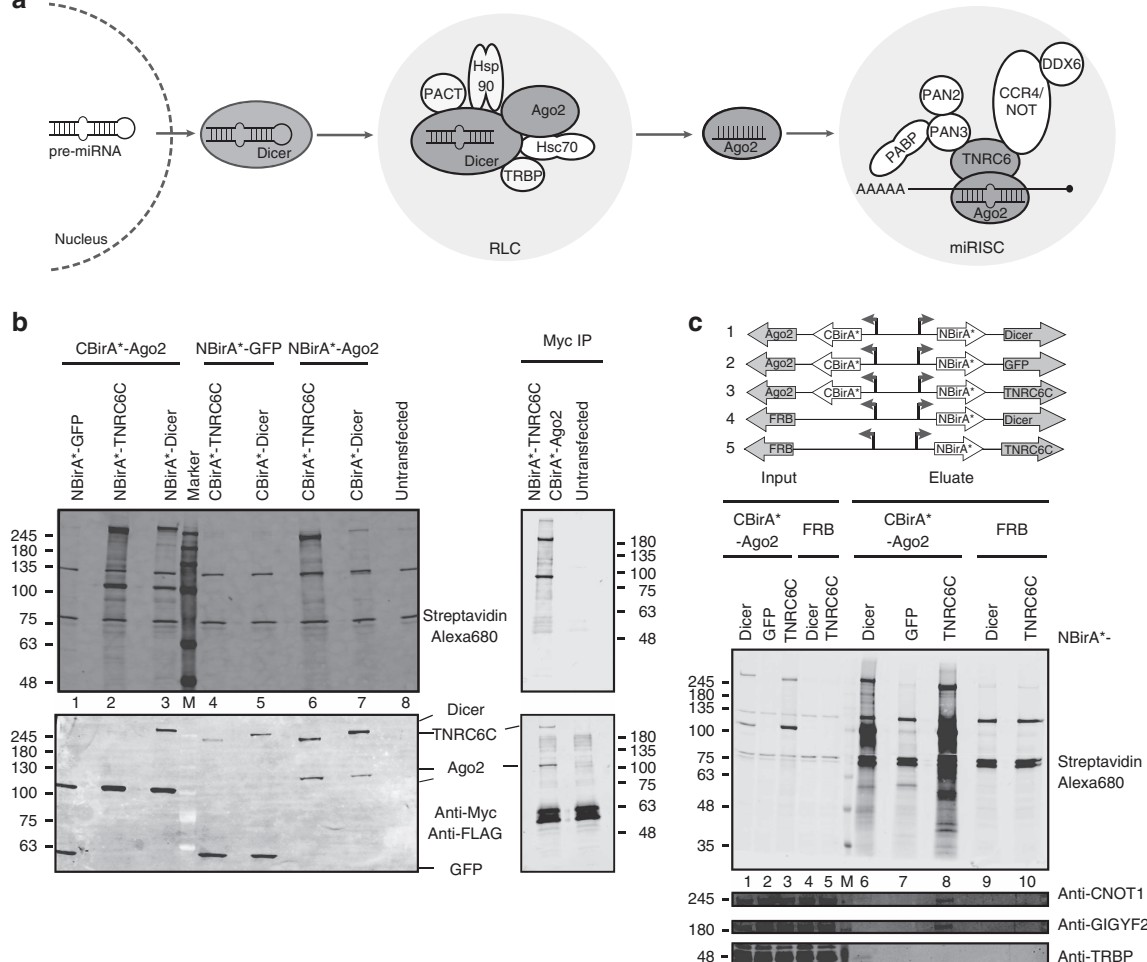

**Figure 5 | Monitoring maturation of the miRISC with split-BioID.** (**a**) Schematic of the miRNA-mediated silencing pathway. Main protein components of the RLC and miRISC are indicated. Dark grey circles are the proteins used for split-BioID. (**b**) Left, blots of lysates of cells transfected with the indicated combinations of fusion proteins. CBirA*-Ago2 combined with NBirA* fused to Dicer, TNRC6C or GFP showed the best interaction-induced biotinylation and was thus used further (See Supplementary Fig. 4). Right, immunoprecipitation of Myc-tagged NBirA*-TNRC6C from transfected or control cells. (**c**) Streptavidin capture of the biotinylated proteins from the CBirA*-Ago2/NBirA*-Dicer, -TNRC6C and -GFP samples. As controls, NBirA*-Dicer and -TNRC6C were also expressed in the absence of CBirA* fragments (replaced by FRB). CNOT1 and GIGYF2 were specifically detected in the Ago2/TNRC6C sample and TRBP in the Ago2/Dicer eluate.

two main hubs (Supplementary Fig. 5). Hub1 is centred on the polyA-binding protein (PABPC) and comprises miRISC components. Hub2 is centred on Hsp90 proteins and comprises RLC components. When combined, the proteins identified from the Ago2/Dicer and Ago2/TNRC6C split-BioID show a similar network of PPI clustering around the same two main hubs (Supplementary Fig. 5a).

When analysed separately, Ago2/TNRC6C-enriched proteins comprised proteins of Hub1, coherent with the functional role of this complex as a regulator of mRNA function. Consistently, gene ontology analysis for these proteins showed that the most enriched terms are poly(A) RNA binding, ribonucleoprotein granule and post-transcriptional regulation of gene expression (Supplementary Fig. 6a). Ago2/TNRC6C split-BioID identified core components of the miRISC that were present (*PABPC*, *DDX6*) or absent (such as subunits of the CCR4/NOT complex) in AP-MS data sets (Fig. 7c). The two other TNRC6 paralogs, A and B, were the most abundant proteins detected by split-BioID, this may reflect the fact that mRNAs often harbour multiple miRNA sites and thus can be bound by multiple miRISC complexes that may come in close proximity, alternatively TNRC6

proteins might bind Ago as multimers. In addition, proteins that regulate miRISC action (such as *RC3H2* (ref. 30), *FMR1* (ref. 31) and *ATXN2L* (ref. 32)) were also identified. Interestingly, an importin (*TNPO1*) was found in this data set, which is in line with data showing that TNRC6 navigates Ago into the nucleus of mammalian cells[33,34]. Additional factors with no previously known direct connection to the miRNA pathway comprise several classes of proteins. Half of these proteins are RNA-binding proteins (RBPs) and thus some of them might be found in proximity to the miRISC because they bind to common target mRNAs. Yet, some of these RBPs are known to directly interact with miRISC-associated factors (for example, PATL1 binds to DDX6 (ref. 35) and HELZ binds to CNOT1 (ref. 36)) and thus may play a role in miRNA-mediated silencing. The other proteins may just reflect the direct proximal cellular environment of the miRISC without necessarily having any role in the miRNA pathway or may represent novel miRISC-associated factors.

Conversely, the Dicer-enriched proteins comprised proteins of the Hub2. The most enriched gene ontology terms are heat shock protein binding, cytosol and protein refolding (Supplementary Fig. 6b). Out of the 12 proteins identified (excluding Dicer and

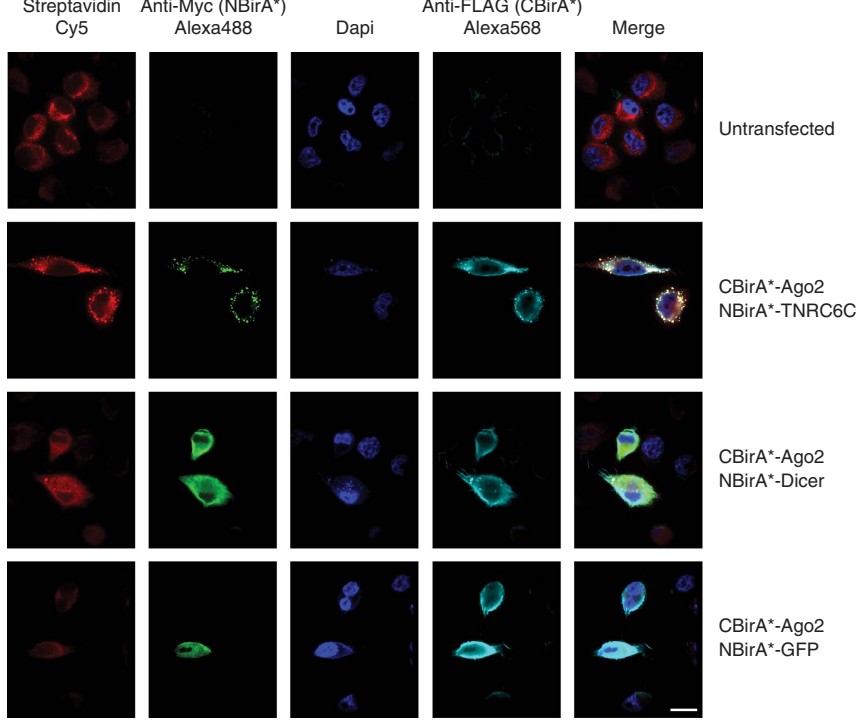

**Figure 6 | Localization of the fusion proteins used for the Cdc25C/14-3-3ε split-BioID.** Immunofluorescence of transiently expressed constructs by Myc- and FLAG-tag detection as well as the detection of biotinylated proteins by Cy5-coupled streptavidin. Scale bar, 15 μm. All constructs are expressed in the cytosol and have the chance to interact with each other.

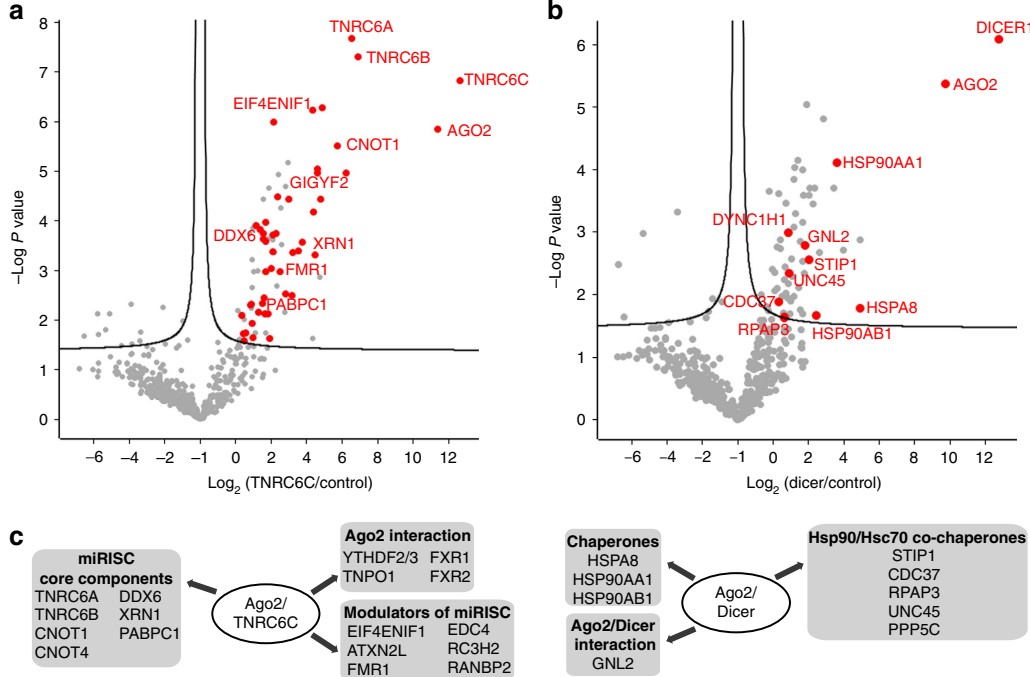

**Figure 7 | Main hits enriched in Ago2/Dicer and Ago2/TNRC6C split-BioID.** Volcano plot showing proteins enriched in the TNRC6C (**a**) or Dicer (**b**) sample over the control BioID samples as in Fig. 3c. Hits that showed higher LFQs than in the GFP and the Dicer samples (**a**) or in the GFP and TNRC6C samples (**b**) are indicated in red. For clarity, not all hits are labelled (**c**) Selected hits from the MS analysis of Ago2/Dicer (right) anf Ago2/TNRC6C (left) split-BioID samples. The complete list is given in Supplementary Data 2.

Ago2), eight are components of the Hsc70/Hsp90 chaperone machinery (Fig. 7c). The Hsc70/Hsp90 machinery was shown to stabilize empty Ago2 (ref. 22) but also to have an evolutionary conserved role in stimulating efficient miRNA loading[37–42].

Hence, by analogy to the role of the same chaperone machinery for the activation of steroid receptors[43], Hsc70/Hsp90, and a cohort of co-chaperones, are proposed to actively participate in the process of miRNA loading into Ago[24]. The identity of the

co-chaperones assisting Hsc70/Hsp90 in loading Ago2 has remained elusive. Split-BioID identifies five of them (Fig. 7b). While Cdc37 (ref. 39) and Hop (*STIP1*)[44] were already shown to affect RISC loading, PP5 (*PPP5C*) was shown to associate with plant Ago1 and Hsp90 (ref. 37). *RPAP3* acts as an Hsp90 co-chaperone within the R2TP complex, which helps the assembly of ribonucleoprotein complexes[45] but a connection with miRISC assembly has not been described yet. The same goes for the last identified co-chaperone, *UNC45A*. Both will be subject of future studies.

Other previously found Ago2-associated chaperones, such as Fkbp4/5 and p23 (refs 38,39), as well as the RLC proteins TRBP[28,29] and PACT[46] were absent from the Dicer-enriched data set. Similarly, not all subunits of the CCR4/NOT complex were found in the TNRC6C-enriched data set. One possible explanation may be that these proteins were out of the labelling range of split-BioID or may not have accessible acceptor lysine residues for biotinylation. Another explanation may be the detection limit of the assay. In the Ago2/Dicer sample, TRBP was detected by western blot. Conversely, PACT and CNOT2 were detected by MS but showed enrichment below our significance threshold. Given the exceptional strength of the streptavidin/biotin interaction, directly performing trypsin digestion on the streptavidin-coupled beads rather than first eluting biotinylated proteins, as we did, may lead to deeper identification coverage. Nonetheless, split-BioID allowed us to specifically probe the proteomes of the miRISC and RLC.

**Identification of a novel miRISC-regulating factor.** We then asked whether split-BioID could identify novel factors associated with the miRNA pathway. As we are interested in novel regulators of the miRISC and the Ago2/TNRC6C split-BioID yielded more unknown proteins, we focused on this data set. We decided to study the protein GIGYF2 as previous data indicated that it may be part of a translation repression complex[47]. GIGYF2 was previously neither found as associated with Ago2 or TNRC6 proteins by AP–MS nor described to play a role in miRNA-mediated repression. We first analysed by western blot, the presence of GIGYF2 in the split-BioID samples. As it was clearly detected in the Ago2/TNRC6C but not in the Ago2/Dicer sample (Fig. 5c), split-BioID assigns GIGYF2 to the miRISC rather than the RLC. Immunofluorescence microscopy was performed to detect endogenous Ago2 and GIGYF2, while Ago2 was detected both in the nucleus and cytosol as previously reported[48], GIGYF2 was mainly found in the cytosol where it partially co-localized with Ago2 (Fig. 8a). To address if Ago or TNRC6 interact with GIGYF2, co-IP experiments were performed. For these experiments, the TNRC6A paralog was studied due to the availability of suitable antibodies. Endogenous Ago2 and TNRC6A could be detected in immunoprecipitation performed with antibodies directed against endogenous GIGYF2 (Fig. 8b). RNAse treatment abolished interaction with Ago2 but a weak signal was still observed with TNRC6A. However, no signal for GIGYF2 was observed in the reverse co-IP performed with an antibody directed against Ago2 while a very faint but reproducible signal was observed in the TNRC6A immunoprecipitation (Fig. 8b). Altogether, these weak signals may explain why GIGYF2 was never found in AP–MS approaches applied to Ago2. TNRC6 proteins harbour a proline-proline-glycine-leucine (PPGL) sequence, which is a typical recognition motif for GYF domains. Accordingly, TNRC6A has been found in pull-down experiments performed with the isolated GYF domain of GIGYF2 fused to GST[49]. To test if the PPGL motif of TNRC6C mediates a direct interaction with the GYF domain of GIGYF2, we expressed recombinant maltose-binding protein (MBP)- and

His$_6$-tagged fragments corresponding to the C-terminal effector domain (CED) of TNRC6C (ref. 6) harbouring the PPGL motif, or an AAGL variant thereof (Fig. 8c). In addition, we expressed two GST-tagged fragments of GIGYF2: (532–740) comprised the GYF domain while (607–740) did not (Fig. 8c). Strikingly, after mixing and immobilization on Ni-NTA beads, MBP-CED-WT-His$_6$ could be co-eluted with GST-GIGYF2 (532–740) comprising the GYF domain of GIGYF2 (Fig. 8d). Binding was considerably reduced on mutation of the PPGL motif to AAGL within the CED, or when the (607–740) fragment of GIGYF2 without GYF domain was used. Altogether, GIGYF2 shows an interaction with Ago and TNRC6, mediated by a direct interaction of the PPGL motif of TNRC6 with the GYF domain of GIGYF2. Within the cellular context, this interaction seems stabilized by RNA and is probably transient as it is well detected by split-BioID but hardly by co-IP. We next tested if GIGYF2 modulates the activity of miRISC. We and others[50–52] have previously shown that miRNA-mediated repression is established through three successive steps: translational repression, followed by mRNA deadenylation, which is tightly coupled to mRNA decay. Previously, we have described inducible cell lines expressing luciferase reporters of miRNA action that allow studying either the translation repression or the mRNA decay component of miRNA-mediated repression. Indeed, when the reporters are analysed 2 h post induction, they are essentially repressed at the translation level, while at later time points mRNA decay mostly accounts for the observed repression[51]. Using a pool of specific siRNAs, GIGYF2 was depleted from these cell lines (Fig. 9), and luciferase activity was analysed at various time points (Fig. 9a). Strikingly, at 2 h post induction, miRNA-mediated repression was alleviated while at later time points repression was unaffected. By contrast a knockdown of the TNRC6 proteins that are necessary for all steps of miRNA-mediated silencing alleviates repression at all time points, while knockdown of subunits of the CCR4-NOT complex (involved in the mRNA destabilization steps and partially in the translation repression steps) also affects the later time points.

Altogether, split-BioID correctly identified GIGYF2 as a miRISC-associated factor and allowed identifying a novel regulator of miRISC activity, at least for the reporter we studied. Our data suggest that GIGYF2 directly and transiently associates with TNRC6 proteins and specifically favours the translational repression component of miRNA-mediated silencing but does not modulate mRNA decay. Interestingly, in zebrafish, a conserved PPGL motif within TNRC6A had been described as mediating translational repression independent of mRNA dead-enylation and decay[53]. This has led to the hypothesis of a yet-to-discover factor that binds to TNRC6 proteins through this motif[54]. Our data reveal that GIGYF2 is that missing factor and further experiments will aim at deciphering its precise mechanism of action. In that vein, a split-BioID applied to the TNRC6/GIGYF2 pair will be of upmost interest.

## Discussion

In summary, we have shown that it is possible to split BirA* in two PCA-suitable fragments. Like other already available PCAs, Split-BioID can be used to validate binary interactions in their native cellular environment. Split-BioID is rather not suited for a high-throughput assay; however, the innovation that singles it out when compared to other PCAs is the possibility to identify additional factors associated with the pair of interacting proteins. As biotinylation of vicinal proteins only happens when and where two proteins interact, split-BioID is a conditional proteomics approach that identifies spatially and temporally defined dynamic complexes. Global proteomics approaches for identifying protein

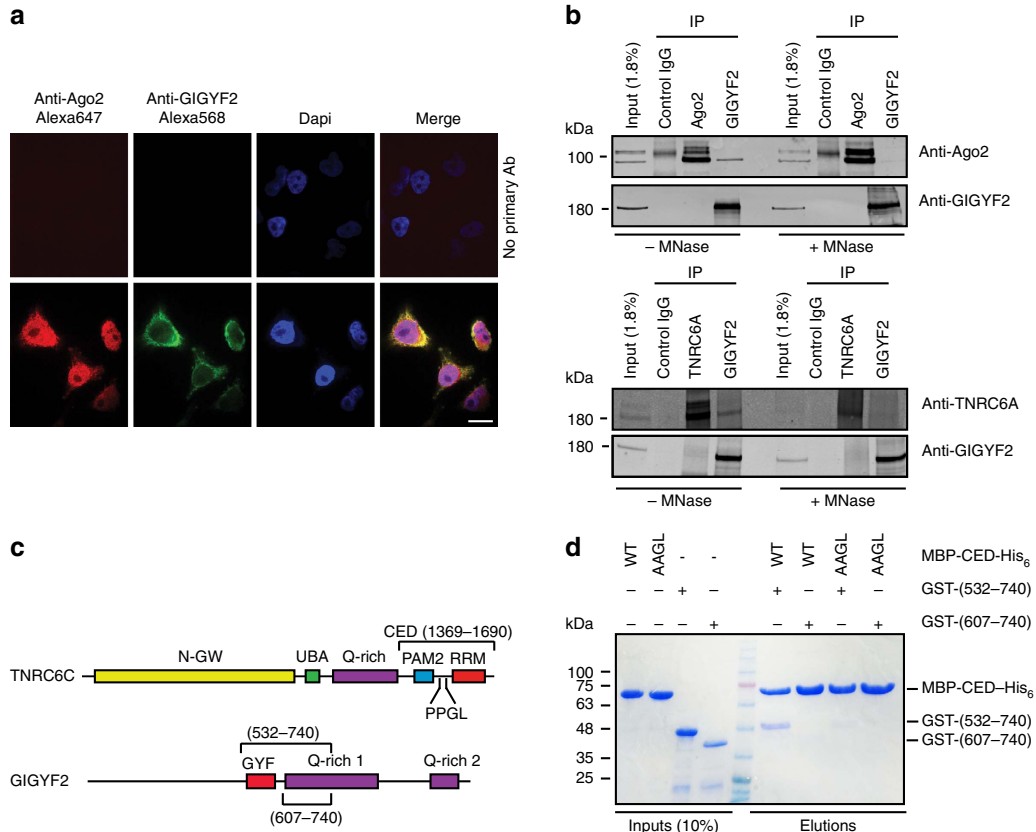

**Figure 8 | Identification of a novel miRISC factor. (a)** Immunofluorescence of endogenous Ago2 and GIGYF2 in HeLa 11ht cells. Scale bar, 15 µm. **(b)** Blots of HeLa cell lysates immunoprecipitated with the indicated antibodies. The blots were decorated with antibodies directed against Ago2, TNRC6A or GIGYF2 as indicated. When indicated ($+/-$ MNase), lysate were pre-treated with micrococcal nuclease. This panel is representative of three different experiments. **(c)** Schematic representation of TNRC6C and GIGYF2 and fragments used for the *in vitro* binding assay. Main domains are highlighted as well as the PPGL motif within TNRC6C. N-GW: N-terminal GW-rich region, UBA: ubiquitin associated-like domain, PAM2: poly(A) binding protein (PABP)-interacting motif 2, RRM: RNA-recognition motif, CED: C-terminal effector motif[6]. **(d)** His$_6$ pulldowns on Ni-NTA beads using recombinant MBP-CED-His$_6$ (WT or AAGL variant) and GST-tagged fragment of GIGYF2 comprising the GYF domain (532–740) or not (607–740).

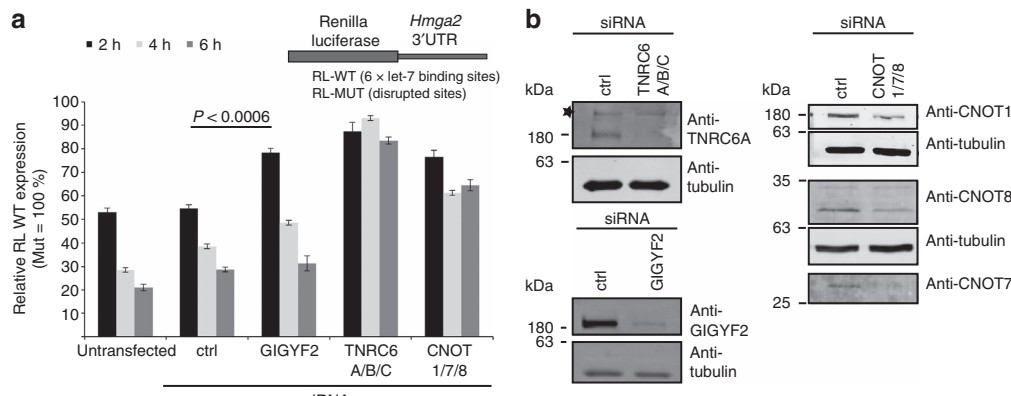

**Figure 9 | Validation of novel miRISC-regulating factor. (a)** Dual luciferase assay following knockdown experiments with the indicated siRNA pools performed in stable inducible cell lines expressing the depicted renilla luciferase reporters. The bars show the expression levels of the RL-WT reporter (with miRNA-binding sites) normalized to a co-expressed control firefly luciferase reporter at the indicated time points after induction. Expression levels are indicated as percentages of values for the matching non-repressed MUT reporter (with no miRNA-binding sites), which are set to 100% at each time point. At 2 h post induction, miRNA-mediated repression is significantly alleviated on knockdown of GIGYF2. *P* value was calculated using a two-tailed student's paired *t*-test ($n = 6$, error bars are s.e.m.). **(b)** Western blots showing knockdown efficiency of the siRNAs used in **a**. Unspecific bands are labelled with a star.

complexes, such as protein correlation profiling[55] or co-elution profiling[56], were successfully applied to get a general overview of cellular protein complexes. These approaches are particularly

strong at identifying complexes with defined localization and composition, however they seem to fail at identifying dynamic complexes as many centrosomal proteins were missed by protein

correlation profiling and co-elution profiling failed to identify the miRISC or CCR4/NOT complexes, possibly because both methods require cell lysis and lysate fractionation. Split-BioID is an ideal complement to these global approaches as it focuses on specific dynamic complexes in their native cellular environment. A potential limitation of the approach comes from the speed of labelling. In BioID, efficient biotinylation has been reported to require 6–24 h. Though the conditional activation of split-BioID ensures it labels proteins involved in temporally defined interactions, proteins with very short half-lives will likely be difficult to detect. Identifying a more active variant of BirA* or designing a PCA based on the engineered peroxidase APEX2 that mediates strong biotinylation within a minute[57] may circumvent this limitation.

Another potential caveat is the level of expression of the fusion proteins. We have co-expressed NBirA* and CBirA* from the same plasmid both under inducible CMV promoters. This has resulted in having one fusion protein being more expressed than its endogenous counterpart. In this study, we focused on establishing the technique and have not explored tuning down the induction of the fusion proteins. However, in future studies, using stable rather than transient transfection is advisable. Indeed, transient transfection may lead to hetero-geneous populations of cells that exhibit a wide range of expression levels. In those cells where the fusion proteins are highly overexpressed interactomes may not faithfully reflect physiological interactions and complicate the interpretation of BioID data. Using stable cell lines for the Cdc25C/14-3-3ε pair led to lower expression of both fusion proteins but CBirA*-Cdc25C was still overexpressed. Rather than a strong discrepancy in the co-expression of both fusion proteins, we believe it probably reflects that Cdc25C is expressed at much lower levels than 14-3-3ε (ref. 58). Hence, if relative low expression of both fusion proteins is needed, one may turn to a dual plasmid system or a single plasmid in which co-expression is driven by two promoters of different strength. Alternatively, tagging through genome editing would ensure physiological expression levels. Of note, as we observed stronger re-activation with our split-BioID assay based on the E256/G257 splitting site than with the alternative E140/Q141 splitting site[12], it will most probably be better suited when low expression levels of the fusion proteins are desired.

As we observed that the extent of biotinylation was depending on which protein was appended to the NBirA* or CBirA* fragment, we recommend testing both iterations when trying split-BioID on a pair of proteins. Similarly, N- and C-terminal fusion may be also tested if the proteins of interest can be tagged on both ends. The long flexible linkers we have used in our assay were taken from another PCA[17] and worked for all the proteins we tested, hence they are a good starting point when testing the assay. To modulate the labelling range of the assay, one may envisage using shorter or longer linkers. However, as for any assay making use of fusion proteins, the length and nature of the linkers may need to be optimized for each protein pair studied.

To demonstrate the usefulness of the technique, we have applied split-BioID to the miRNA silencing pathway. Remarkably, activating the same protein (Ago2) with two different interacting partners (Dicer or TNRC6C) led to the labelling and identification of two very distinct sets of proteins reflecting two different functional steps in the life cycle of Ago2 and miRISC maturation. Importantly, as opposed to AP-MS, split-BioID was able to identify components of the major downstream effectors of Ago such as the CCR4/NOT deadenylation complex. As Ago2 is also reported to act at various defined cellular locations, such as the nucleus, the endoplasmic reticulum or multi-vesicular bodies[59], we plan to exploit the power of split-BioID to analyse the elusive composition of the corresponding protein complexes.

Two other techniques have properties comparable to split-BioID. A conditional proteomics approach was recently used to specifically identify proteins involved in zinc homoeostasis[60]. However, it necessitated the chemical synthesis of a custom $Zn^{2+}$-responsive labelling reagent, which makes the approach difficult to generalize. Moreover, as in AP-MS, the proteins identified are not assigned to specific complexes. A second approach, applied in yeast, proposes a non-radioactive pulse chase epitope labelling for the time-resolved AP of nascent protein complexes[61]. It relies on the incorporation of a non-natural amino acid and a riboswitch-controlled translation arrest. Whether this can be applied to another organism than yeast, notably for the identification of a suitable riboswitch and to obtain sufficient quantities of sample for MS analysis, remains to be addressed. In addition, as the bait must be purified at different time points following chase, the method can only be applied to monitor a relatively slow maturing complex and requires tightly timed handling to capture a specific maturation step. By contrast, split-BioID is activated in the native cellular environment only when two proteins interact, whenever the interaction takes place. With its ability to unbiasedly probe, in a single and simple assay, complexes building around a pair of interacting proteins, split-BioID is thus unique in the toolbox of methods for the analysis of PPI.

Altogether, we present a technique that will complement existing methods for the study of dynamics PPI networks and the assembly of protein complexes. split-BioID is a readily available method that only necessitates standard lab equipment and reagents. With the possibility to control the spatiotemporal activation of BirA*, we expect split-BioID will also be a very valuable tool for the otherwise challenging study of organelle contact sites or the protein composition of dynamic RNA granules, and hence contribute to the definition of high-resolution subcellular maps.

## Methods
**Plasmids and antibody.** The plasmids, primers and antibodies used in this study are provided in the Supplementary Tables 1–5

**Cell culture.** Low passage HELA 11ht cells, a subclonal HeLa-CCL2 cell line, stably expressing the reverse tetracycline-controlled transcription activator rtTA-M2 and containing a locus for Flp-recombinase-mediated cassette exchange[62] were obtained from Dr Kai Schönig (ZI Mannheim). Cells were regularly tested for mycoplasma contamination. Cells were grown in DMEM medium (Sigma) containing 10% tet-free bovine serum (Biowest), 200 µg ml$^{-1}$ HygromycinB (Sigma) and 200 µg ml$^{-1}$ G418 (Sigma).

**Construction of plasmids.** The different split-BioID constructs were designed for co-expression using the pSF3 backbone that harbours a bidirectional tetracycline-responsive promotor[51]. NBirA* and CBirA* fragments were amplified by PCR using a pSF3-BirA* plasmid as a template, and introducing either SalI and AscI or PacI and BglII as cutting sites, respectively. CBirA* fragment amplification was done using the C66, C271 or C274 forward primer together with the BirA* reverse primer. NBirA* fragment was amplified using the BirA* forward primer together with the N65, N270 or N273 reverse primer. FRB was amplified with additional restriction sites for FseI and PacI while FKBP was designed with AscI and BamHI. The glycine–serine linkers were either integrated between FRB and CBirA* using PacI (linker 1) or NBirA* and FKBP using AscI (linker 2) restriction sites.

The optimal split-BioID plasmids (four combinations of E256/G267) were also designed with the pSF3 backbone but CBirA* and NBirA* were ordered as gBlocks (IDT), fused to a sequence coding for the same glycine–serine linkers (QISYASRGGGSSGG and GGGSSGGQISYASRG) that were used in a split luciferase-based PCA[17], and to specific restriction sites to insert the fusion proteins of interest. Most fusion proteins used were either integrated via PmeI and PacI restriction sites or ClaI and MluI depending on the fusion to the CBirA* part or the NBirA* part, respectively. Fusion proteins where these restriction sites could not be used were designed using the NEBuilder tool. All fusion proteins were amplified by PCR.

E140/Q141 construct was designed by replacement of the E256/G267 sBirA* parts in construct 2 with PCR amplified FKBP-NBirA*E140 and CBirA*Q141 using ClaI/BamHI and EcoRI/BglII restriction sites, respectively.

For expression of recombinant proteins, the CED domain of TNRC6C (amino acids 1369–1690) was amplified by PCR and transferred using the restriction sites BamHI and NheI to a modified pMal-c2x plasmid (New England Biolabs) that allows N-terminal tagging with MBP and C-terminal tagging with six histidines. The corresponding AAGL coding mutant was obtained by site-directed mutagenesis. The two fragments of GIGYF2 (amino acids 532–740 and 607–740) were amplified by PCR and transferred using the restriction sites MfeI and NotI into a pGEX-6p plasmid (GE Healthcare) that allows N-terminal tagging with GST.

**Immunoprecipitations.** For each sample, 35 µl slurry of protein-coupled magnetic beads (CST or NEB (25 µl slurry)) was used. Beads were coupled to 2.5–5 µg antibody at 4 °C for 1 h in wash buffer (50 mM Tris pH 7.4, 300 mM NaCl, 1 mM MgCl₂, 0.5% NP-40). Cell lysates were prepared as described above (Screening for biotinylation). Same protein amounts (100–250 µg) were loaded on pre-coupled beads, the samples were then adjusted to 250 µl with lysis buffer, and incubated overnight at 4 °C on a rotating wheel. On the next day, the beads were washed four times with IP wash buffer and elution was done by either boiling (in case of anti-Myc IP) or incubation with FLAG peptide (150 ng µl⁻¹ in TBS) for 30 min at 4 °C. The resulting samples were analysed by western blot.

For co-IP of endogenous proteins HeLa 11ht cells were grown in a 10 cm dish and collected in 1.4 ml lysis buffer, which was incubated for 30 min at 4 °C. Cleared lysates where treated with or without micrococcal nuclease (14 gel units per µl, NEB) and 1.5 mM CaCl₂ for 25 min at room temperature (RT). Samples were afterwards loaded on pre-coupled beads using either anti-Ago2, anti-TNRC6A, anti-GIGYF2 or anti IgG control antibodies (each 5 µg). Incubation, washing and elution were performed as described above.

**Screening for biotinylation.** HeLa 11ht cells were seeded at a concentration of 1 × 10⁵ cells per well of a six-well plate or 8 × 10⁵ cells per 10 cm dish the day before transfection. Transfection was performed using polyethylenimine (Polysciences, Inc.) in a 2:1 (w/w) ratio to the added DNA amount. Plasmid DNA (3 µg for six well and 6 µg for 10 cm), polyethylenimine and DMEM (without serum) were mixed in a total volume of 500 µl, incubated for 5 min at RT and added to the cells. The media of the cells was changed directly before transfection. The day after transfection, biotin (Sigma) was added to the medium (50 µM) and the cells were induced with 200 ng ml⁻¹ doxycycline (Sigma). In addition, for FRB/FKBP constructs, rapamycin (Invivogen) was added to 100 nM. Lysates were prepared 24 h after induction as follow: cells were washed once with PBS and scraped with 100 µl lysis buffer (50 mM Tris pH 7.4, 150 mM NaCl, 2 mM EDTA, 0.5% NP-40, 0.5 mM DTT and complete protease inhibitor (Roche)) followed by a centrifugation step at 14,000g for 10 min at 4 °C. Protein amounts were determined with a Bradford assay (Expedeon) and equal protein amounts were loaded on SDS–PAGE gels and analysed by western blot.

**Western blot analysis.** After SDS–PAGE, proteins were transferred to a low fluorescence PVDF membrane (Millipore) using the high molecular weight programme of the Trans-Blot Turbo Transfer System (BioRad) or overnight transfer (80 mA, 4 °C) in a Mini Trans-Blot cell (BioRad). Membranes were blocked for 1 h in 5% skimmed milk in PBS at RT and then incubated with the appropriate primary antibodies (Supplementary Table 4). After three washes in PBS-T, membranes were incubated with IRDye-coupled secondary antibodies. After two washes in PBS-T, and one wash with PBS, fluorescent signals were detected by scanning the membranes with an Odyssey CLx imaging system (LI-COR). Quantification was performed with the LI-COR Image Studio software according to the distributor's instructions. Relevant parts of the resulting images were cropped, and linear adjustment of lightness and contrast levels were applied to the whole cropped areas to optimize visualization of the bands without compromising the information of the original picture. Full scans of the blots used to assemble the figures are shown on Supplementary Fig. 7.

**Construction of stable cell lines.** For the construction of stable cell lines (Cdc25C/14-3-3ε constructs for split-BioID, and BirA*-Ago2 and BirA*-control proteins for Ago2 and control BioID, the control proteins were Rab11a, Lamp1, TGN38, GRASP65, RHD4 and Sec61β), HeLa 11ht cells that harbour a stably integrated Hygromycin-TK cassette flanked by flippase recognition target (FRT) sites were used. The split-BioID constructs were designed with the pSF3 backbone[51], which also harbours the same FRT sites and are therefore compatible with Flp-recombinase-mediated cassette exchange in HeLa 11ht cells. Cells were seeded at a concentration of 1 × 10⁵ cells per well of a six-well plate on the day before transfection, which was done using Lipofectamine 3000 (LifeTechnologies) following the manufacturer's protocol (cotransfection of BirA* construct and Flp coding plasmid pPGKFLPobpA (addgene 13793)). Cells were transferred to a 10 cm dish 24 h after transfection and 50 µM ganciclovir (Sigma) was added 72 h after transfection to start the selection procedure. Approximately one week later, colonies had formed and could be picked and propagated. Cells were in total treated for at least 3 weeks with ganciclovir.

**BioID.** For each split-BirA* condition, three biological replicates were performed and analysed via MS. In addition, BioID runs were performed with stable cell lines expressing six BirA*-tagged unrelated proteins (Sec61β, RHD4, GRASP65, TGN38, Lamp1 and Rab11a), these six runs were used as controls for the general background of the technique. Three to four 10 cm dishes per condition (Dicer/TNRC6C/Ago2 constructs) were transfected with the appropriate split-BioID constructs as described above. On the next day, the cells from each 10 cm dish were transferred to 15 cm dishes in medium containing 50 µM biotin, and 200 ng ml⁻¹ doxycycline to induce production of the CBirA* and NBirA* fusion proteins. For stable cell lines, 8 × 15 cm (Cdc25/14-3-3ε constructs) or 2 × 15 cm dishes (BirA*-Ago2 and BirA*-controls) per condition were seeded with a cell amount of 2.6 × 10⁶ cells per dish in biotin (50 µM)-containing medium, and directly induced with 200 ng ml⁻¹ (Cdc25/14-3-3ε constructs) or 25 ng ml⁻¹ doxycycline (BirA*-Ago2 and BirA*-controls). Twenty-four hour post induction, cells were washed twice with PBS, and then scraped in 1.5 ml PBS. Cells were then pelleted (1,200g, 5 min), snap frozen in liquid nitrogen and stored at −80 °C. For cell lysis, cell pellets were resuspended in 1 ml lysis buffer (50 mM Tris pH 7.4, 500 mM NaCl, 0.4% SDS, 5 mM EDTA, 1 mM DTT, 1× complete protease inhibitor (Roche)) at RT and then mechanically disrupted by 10 passages through a 25 G needle followed by sonication (Bioruptor plus sonification device (Diagenode), four cycles at high intensity, 30 s per cycle).

After sonication, Triton X-100 concentration was adjusted to 2% and sodium chloride concentration to 150 mM. Lysates were then centrifuged at 4 °C 16,000g for 10 min. Ten per cent of each supernatant was kept as input material and the rest was incubated in equal amounts (3–3.5 mg per sample depending on the experiment) with 200 µl Dynabeads MyOne Streptavidin C1 (Invitrogen, catalogue number 65002) at 4 °C overnight on a rotating wheel. Magnetic beads were equilibrated for 10 min at RT in equilibration buffer (50 mM Tris pH 7–4, 150 mM NaCl, 0.05% Triton X-100, 1 mM DTT) prior use. All washing steps were performed at RT on a rotating wheel each for 8 min. Beads were washed with four different washing buffers each two times. Wash buffer 1 (2% SDS in water), wash buffer 2 (50 mM HEPES pH7.4, 1 mM EDTA, 500 mM NaCl, 1% Triton X-100, 0.1% Na-deoxycholate), wash buffer 3 (10 mM Tris pH 8, 250 mM LiCl, 1 mM EDTA, 0.5% NP-40, 0.5% Na-deoxycholate), wash buffer 4 (50 mM Tris pH 7.4, 50 mM NaCl, 0.1% NP-40). The biotinylated proteins were eluted from the beads by boiling them for 15 min at 98 °C in 30 µl elution buffer (10 mM Tris pH 7.4, 2% SDS, 5% β-mercaptoethanol, 2 mM Biotin). Beads were then immediately removed and the samples stored at −20 °C.

**Sample preparation for MS.** For MS analysis, eluted samples were run on 4–20% RunBlue SDS precast gels (Expedeon) until they migrated 2–3 cm into the gel. The whole lane was sliced after staining with colloidal Coomassie Brilliant Blue G250 (ref. 63), excluding the streptavidin band. The samples were sent for analysis to FingerPrints proteomics (University of Dundee, UK). There, the samples were processed to overnight (16 h) trypsin digestion (Modified Sequencing Grade, Roche). The peptides were extracted from the gel and dried in a SpeedVac (Thermo Scientific). The peptides were then resuspended in 50 µl 1% formic acid, centrifuged and transferred to HPLC vials.

**Liquid chromatography/MS analysis.** The samples were loaded (15 µl injection volume) on an Ultimate 3000 RSLCnano liquid chromatography system (Thermo Scientific, running dual column set-up) coupled to a LTQ OrbiTrap Velos Pro (Thermo Scientific). The peptides were initially trapped on an Acclaim PepMap 100 (C18, 100 µM × 2 cm) trap column, and then separated on an Acclaim PepMap RSLC C18 column (75 µM × 50 cm) followed by a transfer line (20 µM × 50 cm) attached to an easy-spray emitter (7 µM ID) (Thermo Scientific) to the MS via an easy-spray source with temperature set at 50 °C and a source voltage of 2.5 kV. Peptides were resolved in a gradient of acetonitrile in 0.08% formic acid, increasing the percentage of acetonitrile from 2 to 40% within 120 min, and to 98% within an addition 25 min.

Mass spectra were acquired in a data-dependent mode with automatic switching between MS and MS/MS scans using a top 15 method. Full MS scans were acquired in the Orbitrap mass analyzer over an m/z 350–1,800 range with a resolution of 60,000 and a target value of 10⁶ ions. Peptide fragmentation was performed with the collision-induced dissociation mode. MS/MS spectra were acquired with a target value of 5,000 ions. Ion selection threshold was set to 5,000 counts.

**MS data analysis.** Raw MS files were analysed by MaxQuant[64] version 1.5.5.1. MS/MS spectra were searched with the built-in Andromeda search engine against the Uniprot-human database (downloaded in March 2016) to which common contaminants and reverse sequences of all entries had been added. The search included variable modifications of methionine oxidation, N-terminal acetylation and lysine biotinylation, and fixed modification of carbamidomethyl cysteine. Minimal peptide length was set to seven amino acids and a maximum of two miscleavages was allowed. The false discovery rate (FDR) was set to 0.01 for peptide and protein identifications. For comparison between samples, we used LFQ[65] with a minimum of two ratio counts to determine the normalized protein intensity. We activated the 'match between run' option.

The data were then processed using Perseus version 1.5.5.3 (ref. 66). Identifications from the reverse database, common contaminants and proteins only identified through a modification peptide were removed. Label-free intensities were then logarithmized (base 2) and the samples were then grouped according to the replicates with the six BioID runs on unrelated proteins defined as control group. At least two valid values across the three replicates were required for each identified protein. Following the Perseus analysis pipeline, empty values were imputed with random numbers from a normal distribution to simulate low abundance values below the detection limit of the instrument. For each split-BioID condition, a two-sample $t$-test based on permutation-based FDR statistics was then applied (250 permutations; FDR = 0.07; $S_0 = 0.1$) to identify specific hits over control. To further filter the data, we kept the resulting hits that had a median LFQ across the three biological replicates at least twice as high as GFP split-BioID sample.

Protein tables are given as Supplementary Data 1–3.

**Knockdowns.** Knockdown experiments were performed in 96-well plates using previously described stable inducible HeLa cell lines co-expressing a firefly luciferase control reporter and either a renilla luciferase reporter appended to the 3′UTR of the let-7 miRNA target Hmga2 or a mutant thereof with disrupted miRNA-binding sites[51]. Transfection was performed with jetPrime (Polyplus Transfection) transfection reagent and 50 nM of GIGYF2 esiRNA (Sigma), 16.67 nM of each TNRC6A, B and C siRNA, 16.67 nM of each CNOT1, CNOT 7 and 8, or 50 nM of control siRNA (Qiagen, 1027281) following the manufacturer's protocol. The siRNA targeting the TNRC6 proteins and CNOT subunits were previously described[51]. Twenty-four hour after transfection, media was changed and knockdown was analysed 48 h after transfection via western blot and luciferase assay.

**Immunofluorescence.** Immunofluorescence of sBirA* constructs was performed with transient transfected cells detecting the FLAG- and Myc-tag and biotinylated proteins. Cells were fixed with 4% formaldehyde for 15 min at RT followed by three washing steps with PBS each 5 min. Permeabilization was performed with 0.2% Triton X-100 in PBS for 20 min at RT. Before incubation of primary antibodies over night at 4 °C, the cells were blocked in 5% BSA/2% goat serum/2% donkey serum in 0.2% Triton X-100/PBS for 1 h at RT. Next day, cells were washed 4 × with 0.2% Triton X-100/PBS for 5 min. Incubation with secondary antibodies was done for 2 h at RT followed by three washing steps in 0.2% Triton X-100/PBS and one with PBS each 5 min. DAPI staining was done for 20 min and cells were afterwards washed again two times with PBS prior fixation with Fluoromount-G (Southern Biotech). Antibodies and dilutions are listed in Supplementary Table 4. Pictures were taken with the × 100 objective of a Perkin Elmer ERS-6 spinning disk confocal microscope controlled by the Velocity software at the Heidelberg Nikon imaging centre. Minimum and maximum displayed values can be found in Supplementary Table 5. Cutoffs for transfected cells were set against untransfected HeLa 11ht cells for all IF pictures and additionally against cells which had not been treated with primary antibodies (not depicted).

**Expression and purification of recombinant proteins.** GST-tagged GIGYF2 fragments and MBP-His$_6$-tagged TNRC6C fragments were expressed in BL21(DE3) cells (New England Biolabs). Cells were grown in 500 ml LB medium at 37 °C until $OD_{600}$ reached 0.6. Thereafter, isopropyl β-D-1-thiogalactopyranoside was added to 0.1 mM to induce protein expression and the cells were grown overnight at 16 °C. The cells were then collected by centrifugation and resuspended in lysis buffer (for GST fusions: PBS + 1 mM DTT, for MBP-His$_6$ fusions: 50 mM Na$_2$HPO$_4$, pH = 8, 500 mM NaCl, 20 mM imidazole, 0.5% Triton X-100, 1 mM DTT). Cells were lysed using a French press and processed for protein purification. GST fusions were purified on glutathione-coupled beads (glutathione sepharose 4B, GE Healthcare) following the vendor's instructions (washing buffer, PBS, 1 mM DTT; elution buffer, 20 mM Tris pH 8, 150 mM NaCl, 20 mM glutathione, 1 mM DTT). MBP-His$_6$ fusions were first purified on Ni-NTA-coupled beads (Ni sepharose high performance GE Healthcare) by incubation of the lysates for 1 h at 4 °C. Followed by two washing steps with wash buffer 1 (100 mM HEPES, pH = 7.4, 500 mM NaCl, 50 mM imidazole, 0.5% Triton X-100) and one with wash step buffer 2 (100 mM HEPES, pH = 7.4, 500 mM NaCl, 50 mM imidazole, 0.1% Triton X-100). Elution (100 mM HEPES, pH = 7.4, 200 mM NaCl, 250 mM imidazole) was performed for 5 min at 4 °C. The eluted sample was then diluted in column buffer (50 mM Tris-HCl, pH = 7.4, 200 mM NaCl, 1 mM EDTA, 1 mM DTT) and then incubated with 250 μl pre-equilibrated amylose-coupled beads (Amylose resin, New England Biolabs). The beads were washed three times with column buffer and elution performed 0.5 ml elution buffer (column buffer plus 10 mM maltose).

**In vitro binding assay.** MBP-CED (WT or AAGL)-His$_6$ and GST-GIGYF2 fragments (600 pmol each) were mixed together in 100 μl binding buffer (50 mM HEPES, pH = 7.4, 150 mM NaCl, 20 mM imidazole, 0.05% Triton X-100, 1 mM DTT) for 30 min at RT. A measure of 10 μl of Ni-NTA-coupled beads were then added to each reaction and incubation resumed for 30 min. The beads were then washed five times with 400 μl cold binding buffer. Bound material was then eluted with 50 μl binding buffer containing 250 mM imidazole.

**Luciferase assay.** Following doxycycline-mediated induction, cells were lysed using cytoplasmic lysis buffer (50 mM Tris-HCl pH 7.4, 150 mM NaCl, 1 mM EDTA pH 8, 0.5% NP-40) for 20 min at 4 °C. FL and RL activities were measured from the lysates on a Xenius XL microplate luminometer (SAFAS Monaco) using the Dual-Luciferase Reporter Assay System (Promega).

**GO enrichments analysis.** For GO enrichment analysis, we used the VLAD application[67] with the human proteome as background. VLAD uses the hypergeometric test for determining significance. To adjust $P$ values, accounting for multiple testing, VLAD calculates a $q$ value that is based on the concept of the positive false discovery rate.

**STRING interaction networks analysis.** For interaction networks analysis, we used STRING v10.0 (ref. 68) (www.string-db.org), keeping default parameters.

**Statistical analysis.** On Fig. 9, data were tested for normality using the Shapiro–Wilk test. The null hypothesis for this test is normal data. Statistical significances were calculated on the normally distributed data using a two-tailed paired Student's $t$-test. All the compared data had similar variance.

**Data availability.** The MS proteomics data have been deposited to the ProteomeXchange Consortium (http://proteomecentral.proteomexchange.org) via the PRIDE partner repository[69] with the data set identifier PXD005005. The data that support the findings of this study are available from the corresponding author on request.

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

## Acknowledgements

We thank Petra Hubbe for excellent technical support. We thank M. Chekulaeva (Berlin), G. Meister (Regensburg), D. Grimm (Heidelberg), S. Winkler (Nottingham), C. Freund (Berlin) and G. Stoecklin (Mannheim) for providing us with plasmids (see Supplementary Data), and K. Schönig (Mannheim) for the HeLa 11ht cell line. We thank the Ruiz de Almodóvar group (Heidelberg), especially P. Himmels for the help and support with the immunofluorescence and sharing antibodies. We thank F. Wieland, H. Hillen and J. Schopp for comments on the manuscript, Life Science Editors for editing assistance. This work was supported by the excellence initiative of the German research council (DFG-EXC81), I.M.S. and J.B. were partially supported by a collaborative research grant (DFG-SFB638) of the German research council.

## Author contributions

I.M.S. performed the experiments presented in Figs 1–9 and Supplementary Figs 1–7. C.C.A.R. contributed with the Ago2-BioID, co-IPs and GIGYF2 knockdowns (Figs 8b and 9 and Supplementary Fig. 5). J.D. performed the binding assay (Fig. 8d). E.K. and M.S. helped with cloning and testing of the constructs. K.W. suggested the best working split. I.M.S. and J.B. designed the study, analysed the data and wrote the paper. All authors edited the paper.

## Additional information

**Competing interests:** The authors declare no competing financial interests.

