## [Peer Review File · Nature Communications]

Reviewers' Comments:

Reviewer #1 (Remarks to the Author):

The manuscript 'Split-BioID: a conditional proteomics approach to monitor the composition of spatiotemporally defined protein complexes' by Schopp et. al., describes the creation and proof of principle application of binary protein complementation to the BioID system of proximity protein biotinylation. Overall the premise of split-BioID is sound and the major objective of creating a functionally reconstitutable set of BioID fragments was tested. However, the studies that apply this technology do not appear to meet the current standards for BioID studies, specifically there is a lack of some appropriate controls and experiments described below. Once these deficiencies are addressed I can be appropriately enthusiastic about the manuscript.

It appears that fundamental BioID controls were not performed for at least some of the MS/pulldown experiments (CDC25C/14-3-3). At the very least cells expressing BioID alone serve to exclude proteins that non-specifically interact with the biotin affinity purification matrix and/or naturally biotinylated proteins. Along these lines in Fig 2C AHNAK/AHNAK2 and Filamin A are typically among the most abundant background proteins detected with the BioID system. Also the histones may be naturally biotinylated, and regardless are typically found in BioID-only or even pulldowns from cells that don't express any BioID proteins, which is the minimal control, albeit not up to contemporary standards. The presence of these proteins among the results in Fig 2C suggests that proper controls are not being used to exclude background proteins. Also in Supp table 1 ACACA is a naturally biotinylated carboxylase and should be listed with the mitochondrial carboxylases.

From my interpretation of the methods it appears that naturally biotinylated carboxylases were used as standards for thresholding. The use of the naturally biotinylated carboxylases as standards for thresholding seems flawed as the total amount of biotinylated proteins captured in a pulldown is highly variable depending on the level of BioID expression and activity. If little to no biotinylation has occurred then those naturally biotinylated proteins are more readily captured and detected since they are the only ones present, whereas if there is extensive competition from proteins biotinylated by BioID then the MS detection and possibly even capture of those naturally biotinylated proteins is impaired. This makes these proteins unreliable as standards for MS analysis. In our BioID studies we have typically observed an inverse correlation between MS detection of these carboxylases and the extent of promiscuous biotinylation, although this can be unpredictably variable.

There is a lack of immunofluorescence throughout these studies to ascertain the fusion protein localization and extent of colocalization. If the control fragment is not largely in the same cellular compartment as the other fragment then it is not really a good control. This could impact the value of the GFP-splitBioID as a control since it would be expected to be predominantly nuclear whereas the other fusion proteins may be predominantly cytoplasmic.

Reviewer #2 (Remarks to the Author):

The authors describe the integration of a classical PPI approach, the protein complementation assay, with a recently developed MS-based proximity labeling approach to study dynamic complexes. The authors raise the issue that AP-MS, like most of the PPI approaches, cannot differentiate the association of a target protein with multiple complexes, an aspect which is currently underexplored in PPI studies. There is indeed a clear need for high resolution approaches to address this issue. The technology presented here provides a clear step towards this.

The authors describe the sequential development and exploration of the split BioID concept using different interactions starting from the rapamycin-dependent interaction between FRB and FKBP. They then show the phosphorylation-dependent interaction between Cdc25C and 14-3-3 ϵ , and finally explore the dynamics of Ago2 by fusing the other part of the split BioID to two proteins known to be present in different Ago2 complexes (Dicer and TNR6C). One of the interactions is validated (GIGY2 in the TNR6 complex) using a translation repression assay developed specifically to address RISC biology. The manuscript is well written and the rationale is well structured and easy to follow.

I do have a few concerns around the manuscript as it is presented now:

It is becoming widely accepted that overexpression can be an issue in the study of protein complexes. Also with BioID overexpression of a bait can lead to artifacts. The authors use an inducible system for the expression of the two partners, but do not really address the expression levels in relation to the endogenous levels. In addition, it is unclear to me at the moment whether they used stable cell lines for all the interactions, or just for phosphorylation-dependent interaction. The issue with higher expression levels is important in protein complementation assays as the split parts still have affinity for each other. This may become an issue with higher levels, as maybe supported by the GFP problem described in the manuscript. The authors should also comment on the amount of material (number of cells or dishes) needed for these lower expression levels.

The second important issue relates to the MS data analysis. It is clear that analysis of purification data is challenging but in the meantime there are tools available to address this problem in a decent way. In my understanding, the authors use two independent biological repeats and averaged MaxQuant LFQ values to find the specific interactors using a cut-off based on known background proteins. While this approach can give an idea about the interactomes, it would be more convincing to use other available options (or possibly in parallel). The free PERSEUS package downstream from MaxQuant can easily generate volcano plots with p-values and difference in protein levels on the axis. This will also include statistics in the analysis (containing FDR values). Other approaches could be used as well (SAINT or similar recent tools) although these rely typically on other metrics (e.g. spectral counts). Most of these approaches do require more repeat experiments (typically at least 3), and I also believe that it would be good to have more repeat experiments to make the data more convincing.

A last issue concerns the presence of the chaperone complex in the analysis for Ago2-DICER. While the authors do provide a plausible explanation for the presence of this complex, and there is some supporting evidence in the literature around Ago2, it does raise the question on the specificity of this complex for Ago2-Dicer. Many of these chaperone components can be picked up with other proteins (albeit with classical AP-MS), especially with kinases (cdc37). They are also present at high levels in the cells (can result in false positives, especially for highly expressed proteins – see higher). The statistical analysis that I asked for in my previous comment can support the specificity of the association. If the specificity is shown, the authors should then comment on the possibility that BioID reveals the classical folding pathway required for bait folding as the (reconstituted) BirA* module may be present during the folding process.

For the analysis of dynamic protein complexes, the authors should also address split APEX2 (Ting lab) and co-elution profiling (Emili-Marcotte labs) in their discussion.

Can the authors also comment on the required steps/efforts to optimize the split BioID approach? Is it required to test all possible iterations of N- and C-terminal fusions for every pair? How many linkers should be tested? Can a higher throughput platform be envisioned or is this not possible based on these considerations?

The endogenous reverse co-IP data for a very weak association between GIGY2 and TNRC6A is not really convincing. How many times was this experiment performed? In fact, please provide clear indications on how many times all experiments were performed. Provide also some MW markers on the blots as a reference for the readers.

Line 223-224: rephrase sentence

Line 312-313: rephrase sentence

Typos in figure 3b labels 'untransfected'

Reviewer #3 (Remarks to the Author):

This is a straightforward and interesting manuscript that describes a novel approach to investigate protein-protein interactions in living cells. The manuscript is of broad interest. Additionally, the manuscript provides novel information in the specific field of RNA silencing.

The manuscript is clearly written and the methods are described with sufficient details.

The only comment is that the authors could be more generous with the references and in citing the original relevant literature. The references are there but are not frequently cited. There are many statements based on previous work for which no reference is given.

Reviewer #4 (Remarks to the Author):

Schopp et al.

Split-BioID: a conditional proteomics approach to monitor the composition of spatiotemporally defined protein complexes

The authors have engineered the BioID method, in which a protein of interest is fused to BirA that biotinylates proteins in its vicinity. Subsequent streptavidin pull downs allow for the identification of protein interactors. One advantage is that also transient interactions might be identified with such a method. Here, the authors have split BirA in two fragments that gain activity when fused to two interaction partners (Split-BioID). They validate their method in three systems. First, fusion of the BirA fragments to FKBP and FRB demonstrates that such a system is functional (the interaction can be induced by addition of rapamycin). Second, the authors probe the known interaction between CDC25C and 14-3-3epsilon and identify a number of proteins that play roles in a similar functional context. Third, they analyze human gene silencing complexes. It is known that Ago proteins engage in different interactions and thus the authors reasoned that this system would be ideal for Split-BioID. They use Dicer (to analyze interactions during RISC loading) and TNRC6C (to study factors involved in later stages of gene silencing). The method appears to work well since many known interactors are found and the RISC loading machinery clusters around chaperones and co-chaperones. The gene silencing interactions identify a number of well-characterized interactions deadenylases, DDX6 or PABPC. Finally, the authors chose the identified protein GIGYF2 and characterized it further. They find that it interacts with Ago2 and TNRC6A in co-IPs and knock down leads to effects on reporter genes

that are under miRNA regulation.

The identification of protein interactors and the definition of complex compositions in space and time are essential for the understanding of cellular functions. The development of Split-BioID is an elegant approach to study such interactions and it allows for the discrimination of different complexes forming around a protein of interest. It may also increase specificity. The manuscript is well written and the results are presented clearly. However, the study generally lacks clearer validation experiments. Several other points are listed below.

1. Figure 1F: the Split-BioID has a much lower activity compared to the non-split protein (2.5%). The authors should state whether this could be a problem or even an advantage since background signals are reduced as well. More importantly, the biotinylation pattern in Figure 1F looks different in the two lanes. Does that mean that the two systems produce different interactions? This would be rather problematic.

2. Generally, the Figures are not very well explained in the main text as well as the legends. This is particularly true for Figure 4A. It is not clear what is really done in this co-IP. This needs to be indicated better.

3. Last sentence of the CDC25C chapter: the final statement is unclear. In *Xenopus*, the phosphorylated form binds both 14-3-3 proteins ...the selectivity of these two 14-3-3 isoforms is conserved in humans...? It sounds like that there is no selectivity in *Xenopus*?

4. The same chapter: change ...dissociation constant 57nm.. to ...dissociation constant 57nM...

5. Figure 2A: wt CDC25C is much weaker in the GFP experiment. This could lead to apparently less background, which might not be reality. GFP should be indicated better in the Figure. It obviously migrates at almost the same position as 14-3-3. Maybe add the GFP labeling to the right site of the blot.

6. Please cite Haase et al. when introducing TRBP.

7. Page 9: the statement of regulation of miRISC by IGF2BP1, FMR1 and ATXN2L lacks references, which should be added.

8. The CDC25C interactome is rather descriptive and mainly correlated to similar functions of candidates or published literature. Since the selectivity of the method is analyzed here, it would be important to at least validate these interactors by additional independent approaches. Otherwise it is difficult to estimate whether the presented interactors are real or not. This needs to be added.

9. The validation of GIGYF2 is rather weak. The authors show weak interaction in co-IPs and a potential role in gene silencing is documented only by using one specific luciferase reporter. The effects are also not very strong. I understand that the main aim of the study is the development of Split-BioID but if the authors claim that they have identified a novel gene-silencing factor, they need to characterize it sufficiently. Localization studies could be added to show that the proteins are at least in the same compartments. Several positive controls should be added to Figure 4B to view the observed effects in the context of known gene silencing factors.

10. The authors state that GIGYF2 is part of a translation repression complex. I understand that 4EHP is not found in the Split-BioID data? The authors should nevertheless test in co-IPs whether this complex is involved in miRNA-guided translational repression. These factors should also be tested in the luciferase experiments shown in Figure 4B.

11. Figure 3D: BirA is split between Ago2 and TNRC6C, i.e. only proteins that are in proximity of an Ago2-TNRC6C complex will be biotinylated. Mechanistically, it is difficult to understand why TNRC6A and B are among the identified targets. The authors should at least present a model in which such interactions would make sense.

Dear reviewers,

We would first like to thank you for your insightful comments. We have made substantial revisions to our manuscript to address the raised issues.

We'd like to add the following note:

During revision of this manuscript, another study by De Munter et al, submitted on the same day than ours, was published in FEBS letters and describes a similar assay to ours using an alternative splitting site within BirA. As it appeared from this study that this alternative splitting site yields a weak activity upon reactivation, we carried out a side-by-side comparison of both assays and found that ours performs better as it yields much stronger biotinylation at similar expression levels of the fragments (new Supplemental Fig. 2)

Below, find a point-by-point answer to your comments.

Reviewers' comments:

Reviewer #1 (Remarks to the Author):

The manuscript 'Split-BioID: a conditional proteomics approach to monitor the composition of spatiotemporally defined protein complexes' by Schopp et. al., describes the creation and proof of principle application of binary protein complementation to the BioID system of proximity protein biotinylation. Overall the premise of split-BioID is sound and the major objective of creating a functionally reconstitutable set of BioID fragments was tested. However, the studies that apply this technology do not appear to meet the current standards for BioID studies, specifically there is a lack of some appropriate controls and experiments described below. Once these deficiencies are addressed I can be appropriately enthusiastic about the manuscript.

It appears that fundamental BioID controls were not performed for at least some of the MS/pulldown experiments (CDC25C/14-3-3). At the very least cells expressing BioID alone serve to exclude proteins that non-specifically interact with the biotin affinity purification matrix and/or naturally biotinylated proteins. Along these lines in Fig 2C AHNAK/AHNAK2 and Filamin A are typically among the most abundant background proteins detected with the BioID system. Also the histones may be naturally biotinylated, and regardless are typically found in BioID-only or even pulldowns from cells that don't express any BioID proteins, which is the minimal control, albeit not up to contemporary standards. The presence of these proteins among the results in Fig 2C suggests that proper controls are not being used to exclude background proteins. Also in Supp table 1 ACACA is a naturally biotinylated carboxylase and should be listed with the mitochondrial carboxylases.

Thank you for the comment. We apologize for the misunderstanding of Suppl. Table 1, there we listed the proteins that were detected in either the CDC25C-WT/14-3-3 experiment or CDC25C-S216A/14-3-3 and are hence that were not filtered yet for background. The proteins selected for (old) Fig. 2C were

the proteins showed a LFQ ratio (WT over mutant) higher than 3 times the levels of endogenous biotinylated enzymes. Applying the threshold, most common contaminants such as histones were removed since they were detected in both WT and S216A datasets.

We acknowledge the content of Supp Table 1 may be misleading and now only list the hits after filtering.

Moreover, we acknowledge the concerns of the reviewer on the use of additional controls and endogenous biotinylated proteins as a reference for thresholding (see next point) and therefore completely updated our analysis pipeline (see response to next concern and response to reviewer #2), this efficiently removed common contaminants of BioID experiments. As a consequence the list of proteins discovered for the Cdc25C/14-3-3 experiment changed to some extent and has been updated in Supp table 1 and Fig. 2.

From my interpretation of the methods it appears that naturally biotinylated carboxylases were used as standards for thresholding. The use of the naturally biotinylated carboxylases as standards for thresholding seems flawed as the total amount of biotinylated proteins captured in a pulldown is highly variable depending on the level of BioID expression and activity. If little to no biotinylation has occurred then those naturally biotinylated proteins are more readily captured and detected since they are the only ones present, whereas if there is extensive competition from proteins biotinylated by BioID then the MS detection and possibly even capture of those naturally biotinylated proteins is impaired. This makes these proteins unreliable as standards for MS analysis. In our BioID studies we have typically observed an inverse correlation between MS detection of these carboxylases and the extent of promiscuous biotinylation, although this can be unpredictably variable.

Thank you for pointing out this potential caveat. To address this point and the concern of reviewer #2 on our analysis of the data we have completely changed our analysis pipeline and used the Perseus package to analyze the data. As a control for the data, BioID experiments on 6 unrelated proteins were used. Volcano plots were used to define positive hits independent of the levels of the endogenous biotinylated proteins.

There is a lack of immunofluorescence throughout these studies to ascertain the fusion protein localization and extent of colocalization. If the control fragment is not largely in the same cellular compartment as the other fragment then it is not really a good control. This could impact the value of the GFP-splitBioID as a control since it would be expected to predominantly nuclear whereas the other fusion proteins may be predominantly cytoplasmic.

Thank you for this very good point. We have now included IF data for all our constructs (Fig. 2 and 3). We would like to point out that all the proteins used in this study are known to shuttle between nucleus and cytoplasm.

Reviewer #2 (Remarks to the Author):

The authors describe the integration of a classical PPI approach, the protein complementation assay, with a recently developed MS-based proximity labeling approach to study dynamic complexes. The authors raise the issue that AP-MS, like most of the PPI approaches, cannot differentiate the association of a target protein with multiple complexes, an aspect which is currently underexplored in PPI studies. There is indeed a clear need for high resolution approaches to address this issue. The technology presented here provides a clear step towards this.

The authors describe the sequential development and exploration of the split BioID concept using different interactions starting from the rapamycin-dependent interaction between FRB and FKBP. They then show the phosphorylation-dependent interaction between Cdc25C and 14-3-3 σ , and finally explore the dynamics of Ago2 by fusing the other part of the split BioID to two proteins known to be present in different Ago2 complexes (Dicer and TNRC6C).

One of the interactions is validated (GIGY2 in the TNRC6 complex) using a translation repression assay developed specifically to address RISC biology. The manuscript is well written and the rationale is well structured and easy to follow.

I do have a few concerns around the manuscript as it is presented now: It is becoming widely accepted that overexpression can be an issue in the study of protein complexes. Also with BioID overexpression of a bait can lead to artifacts. The authors use an inducible system for the expression of the two partners, but do not really address the expression levels in relation to the endogenous levels. In addition, it is unclear to me at the moment whether they used stable cell lines for all the interactions, or just for phosphorylation-dependent interaction. The issue with higher expression levels is important in protein complementation assays as the split parts still have affinity for each other. This may become an issue with higher levels, as maybe supported by the GFP problem described in the manuscript. The authors should also comment on the amount of material (number of cells or dishes) needed for these lower expression levels.

Thank you for raising this important point. We have now included data showing the expression levels of all fusion proteins compared with their endogenous counterparts (supplemental Fig. 3 and 4). Unfortunately the comparison could not be performed for TNRC6C as it appears that the full length TNRC6C paralog is not expressed at detectable levels in HeLa cells. This is in line with validation data provided by the antibody vendor where no signal for full length TNRC6C is seen in HeLa lysates, and with functional experiments performed by the Chekulaeva lab using HeLa-11ht cells coming from our lab that showed that knocking-down the TNRC6A and B paralogs is sufficient to completely alleviate miRNA-mediated repression (Mauri M et al, Nucleic Acids Res. 2017). For all the pairs we tested one fusion is overexpressed when compared to its endogenous counterpart while the other fusion is either expressed at comparable or lower levels. We believe the difference has more to do with the expression level of the endogenous

proteins than to a much higher expression of one fusion protein than the other. Indeed, in one case the NBirA* fusion (NBirA*-Dicer) is overexpressed while in the other case the CBirA* fusion (CBirA*-Cdc25) is. Moreover, we know from previous qPCR data on luciferase reporters expressed from the same plasmid (Béthune et al, EMBO Reports 2012) that our dual expression plasmid normally results in comparable expression of the two co-expressed genes.

Since we concentrated here on establishing the assay we have not explored this further (and the MS data identified proteins that are functionally relevant) but we have commented this potential issues and ways to overcome them (Page 10, lines 422-433)

All experiments were performed with transient cell lines, only the MS data for the Cdc25/14-3-3 datasets were obtained from stable cell line. The only reason for this is that we initially were not sure how much we would have to upscale the assay to get sufficient material for MS analysis. It later turned out that transient transfection is completely fine. To make it clear when we used transient or stable cell line we have indicated it in main text, figure legends and methods. Number of dishes, exact transfection conditions, and amount of protein loaded on the streptavidin-coupled beads are described in the methods part (page 14, lines 609-630)

The second important issue relates to the MS data analysis. It is clear that analysis of purification data is challenging but in the meantime there are tools available to address this problem in a decent way. In my understanding, the authors use two independent biological repeats and averaged MaxQuant LFQ values to find the specific interactors using a cut-off based on known background proteins. While this approach can give an idea about the interactomes, it would be more convincing to use other available options (or possibly in parallel). The free PERSEUS package downstream from MaxQuant can easily generate volcano plots with p-values and difference in protein levels on the axis. This will also include statistics in the analysis (containing FDR values). Other approaches could be used as well (SAINT or similar recent tools) although these rely typically on other metrics (e.g. spectral counts). Most of these approaches do require more repeat experiments (typically at least 3), and I also believe that it would be good to have more repeat experiments to make the data more convincing.

Thank you for this comment that we believe helped us improve the quality of our data. We have performed more biological replicates and analyzed the data with the Perseus package. All the datasets now correspond to three biological replicates. For the general background of the method, we performed BioID experiments on 6 unrelated proteins and used the corresponding data as a negative control. We provide the volcano plots and FDR values as requested (new supplemental figure 5). A final further filter was applied as we only kept the hits identified by Perseus that were two fold more abundant than in the GFP split-BioID control. Compared to our original analysis, the datasets changed to some extent but without affecting the main conclusions of the submitted manuscript. As a result Fig 2 and new Fig. 4,

and the supplementary tables were updated.

A last issue concerns the presence of the chaperone complex in the analysis for Ago2-DICER. While the authors do provide a plausible explanation for the presence of this complex, and there is some supporting evidence in the literature around Ago2, it does raise the question on the specificity of this complex for Ago2-Dicer. Many of these chaperone components can be picked up with other proteins (albeit with classical AP-MS), especially with kinases (cdc37). They are also present at high levels in the cells (can result in false positives, especially for highly expressed proteins – see higher). The statistical analysis that I asked for in my previous comment can support the specificity of the association. If the specificity is shown, the authors should then comment on the possibility that BioID reveals the classical folding pathway required for bait folding as the (reconstituted) BirA* module may be present during the folding process.

With the new analysis pipeline we can show that the Hsp90/Hsc70 machinery is specifically associated with the Ago2/Dicer pair. Numerous papers, using mammalian and fly cells or in vitro reconstitution systems, plants and tetrahymena have shown that empty Ago2 needs to be stabilized by Hsp90 and that the Hsp90/Hsc70 assist the loading of miRNAs into Ago2 (references are cited on page 7, line 298). Hence we believe that the identification of the Hsp90/Hsc70 machinery in the Ago2/Dicer split-BioID dataset was expected and is in line with current models.

For the analysis of dynamic protein complexes, the authors should also address split APEX2 (Ting lab) and co-elution profiling (Emili-Marcotte labs) in their discussion.

We added a discussion on the co-elution profiling method (page 9, lines 405-414). To our knowledge no published or pre-print manuscript describes a split-APEX2 approach but we added a comment as to the potential of such an assay (page 10, lines 418-421)

Can the authors also comment on the required steps/efforts to optimize the split BioID approach? Is it required to test all possible iterations of N- and C-terminal fusions for every pair? How many linkers should be tested? Can a higher throughput platform be envisioned or is this not possible based on these considerations?

Generally we would recommend following the steps we performed to define the fusion proteins to use for the Ago2/Dicer/TNRC6 interactions as described in the main text. We have added additional comments/guidelines (page 10, line 434-443). To summarize: as in any assay that relies on tagging proteins, one would need to test the orientation of all fusion proteins if no data is available (like e.g. description of a GFP-tagged version of the proteins of interest). As we have shown (Fig. 3 and S2), it is better to test which protein to fused to NBir* or CBir* to identify the most active combination. The linkers we chose are long and flexible, and hence may be useful for many different

proteins, but of course some proteins may work better with other linkers. Due to the amount of material needed for MS analysis, a higher throughput platform does not seem realistic at this stage. When it come to validate binary interactions, classical PCAs based on split-GFP or split-luciferase are more suitable, the strength and uniqueness of our assay is not the possibility for higher throughput screening but the ability to identify additional proteins associated with a pair of interacting proteins. We have also added this point in the discussion.

The endogenous reverse co-IP data for a very weak association between GIGY2 and TNRC6A is not really convincing. How many times was this experiment performed? In fact, please provide clear indications on how many times all experiments were performed. Provide also some MW markers on the blots as a reference for the readers.

Thank you for this comment and we apologize that the information was not indicated clearly in the manuscript. Apart from the control blots for knock-down efficiency that were performed only twice as effect were consistent in the luciferase assay, all the pictures depicting blots represent one representative picture from at least three independent experiments (biological replicates), this is now mentioned in the figure legends for all experiments. We have also added the MW markers.

We agree that the co-IPs show only very weak signals that, by themselves, would not be in strong support for an interaction between TNRC6/Ago and GIGYF2. We have performed some more co-IP using more starting material but always end up with similar outcome. This may reflect a transient interaction between GIGYF2 and TNRC6 that is hard to capture by co-IP. However, GIGYF2 was clearly detected in both the BioID and split-BioID datasets.

In fact, we would like to stress that we have chosen to further characterize GIGYF2 as an example of a protein that was not detected in IP-MS but consistently showed-up in BioID and split-BioID. Hence it is not completely surprising that signals in co-IP are weak.

To strengthen the hypothesis that GIGYF2 and TNRC6C do interact together through the PPGL motif of TNRC6C, we have performed an in vitro binding assay in which a recombinant domain of TNRC6C with WT or mutated PPGL motif was mixed with a recombinant domains of GIGYF2 containing the GYF domain or not. In this assay, a direct PPGL- and GYF domain-mediated interaction between the two domains could be clearly shown (new Fig. 5c and 5d).

Line 223-224: rephrase sentence (now 209-212)

Line 312-313: rephrase sentence (now 291)

We rephrased both sentence and hope they are clearer now.

Typos in figure 3b labels 'untransfected'

Thank you for your careful review of the manuscript. We have corrected the typos in the aforementioned figure.

Reviewer #3 (Remarks to the Author):

This is a straightforward and interesting manuscript that describes a novel approach to investigate protein-protein interactions in living cells. The manuscript is of broad interest. Additionally, the manuscript provides novel information in the specific field of RNA silencing.

The manuscript is clearly written and the methods are described with sufficient details.

The only comment is that the authors could be more generous with the references and in citing the original relevant literature. The references are there but are not frequently cited. There are many statements based on previous work for which no reference is given.

Thank you for your comments. We are sorry if we gave the impression not to give justice to previous literature. We have quoted the references more often and added more references, notably the ones we had in the supplementary table 2 & 3 but that did not appear in the main text.

Reviewer #4 (Remarks to the Author):

Schopp et al.

Split-BioID: a conditional proteomics approach to monitor the composition of spatiotemporally defined protein complexes

The authors have engineered the BioID method, in which a protein of interest is fused to BirA that biotinylates proteins in its vicinity. Subsequent streptavidin pull downs allow for the identification of protein interactors. One advantage is that also transient interactions might be identified with such a method. Here, the authors have split BirA in two fragments that gain activity when fused to two interaction partners (Split-BioID). They validate their method in three systems. First, fusion of the BirA fragments to FKBP and FRB demonstrates that such a system is functional (the interaction can be induced by addition of rapamycin). Second, the authors probe the known interaction between CDC25C and 14-3-3epsilon and identify a number of proteins that play roles in a similar functional context. Third, they analyze human gene silencing complexes. It is known that Ago proteins engage in different interactions and thus the authors reasoned that this system would be ideal for Split-BioID. They use Dicer (to analyze interactions during RISC loading) and TNRC6C (to study factors involved in later stages of gene silencing). The method appears to work well since many known interactors are found and the RISC loading machinery clusters around chaperones and co-chaperones. The gene silencing interactions identify a number of well-characterized interactions deadenylases, DDX6 or PABPC. Finally, the authors chose the identified

protein GIGYF2 and characterized it further. They find that it interacts with Ago2 and TNRC6A in co-IPs and knock down leads to effects on reporter genes that are under miRNA regulation.

The identification of protein interactors and the definition of complex compositions in space and time are essential for the understanding of cellular functions. The development of Split-BioID is an elegant approach to study such interactions and it allows for the discrimination of different complexes forming around a protein of interest. It may also increase specificity. The manuscript is well written and the results are presented clearly. However, the study generally lacks clearer validation experiments. Several other points are listed below.

1. Figure 1F: the Split-BioID has a much lower activity compared to the non-split protein (2.5%). The authors should state whether this could be a problem or even an advantage since background signals are reduced as well. More importantly, the biotinylation pattern in Figure 1F looks different in the two lanes. Does that mean that the two systems produce different interactions? This would be rather problematic.

Thank you for your comment. We have commented on the lower activity of split-BioID, which in our hand was sufficient to perform BioID experiments. We do not believe that a lower activity is necessarily an advantage as classical BioID with its 40 times higher activity than split-BioID has been applied successfully by many group, choosing appropriate negative controls being probably the most important parameter when dealing with background.

The biotinylation patterns on Fig. 1F can not be compared directly for the two following reasons:

First, we had to load much less protein amount (0.25 μ g vs. 10 μ g as indicated in the figure) for the BioID sample so that the biotinylation signal are in the same non-saturated range to ensure quantifications are correct. This explains why the major endogenously biotinylated proteins are much less visible in the BioID sample, than in the split-BioID and untransfected samples.

Second, in BioID experiments one of the most biotinylated proteins is typically the BioID fusion protein itself. In the case of the BioID sample, this would be the BirA* enzyme (ca. 36 kDa), in the case of the split-BioID sample, this would be the two fusion proteins (CBirA*-FRB and NBirA*-FKBP, 24kDa and 43 kDa respectively). This explains why the major band observed in the BioID sample (corresponding to BirA*) is not present in the split-BioID sample.

2. Generally, the Figures are not very well explained in the main text as well as the legends. This is particularly true for Figure 4A. It is not clear what is really done in this co-IP. This needs to be indicated better.

Thank you for pointing this, it is important for us that readers understand our data. We have added more details in the figure legends and main text so that experiments are more clearly described.

3. Last sentence of the CDC25C chapter: the final statement is unclear. In

Xenopus, the phosphorylated form binds both 14-3-3 proteins ...the selectivity of these two 14-3-3 isoforms is conserved in humans...? It sounds like that there is no selectivity in Xenopus?

The hits for CDC25C changed with adding more repeats and using the Perseus package for analysis. As a consequence 14-3-3zeta is not found anymore so we removed this part of the discussion.

4. The same chapter: change ...dissociation constant 57nm... to ...dissociation constant 57nM...

Thank you for your careful review of our manuscript, we have change the sentence accordingly.

5. Figure 2A: wt CDC25C is much weaker in the GFP experiment. This could lead to apparently less background, which might not be reality. GFP should be indicated better in the Figure. It obviously migrates at almost the same position as 14-3-3. Maybe add the GFP labeling to the right site of the blot.

Thank you for this very good point, we initially had the very same concern but then noticed that the level of expression of the fusion proteins is not necessarily correlated to the biotinylation levels. Indeed on Fig 3B, one can see a clear NBir-GFP signal (lane 1, lower blot) that does not lead to significant biotinylation, while NBir-TNRC6C (lane 2, lower blot) is hardly detectable but leads to strong biotinylation upon association with CBir-Ago2. For MS analysis, our new analysis pipeline compares only conditions in which biotinylation is observed (the general background is deduced from BioID datasets obtained from 6 unrelated proteins), the GFP split-BioID control is only used as a final filtering step to further reduce the number of hits. Hence even if the GFP split-BioID control would yield an artificially low background this would not affect the quality of the MS analysis.

We have followed your suggestion for the labeling of Fig. 2A.

6. Please cite Haase et al. when introducing TRBP.

We have updated the references.

7. Page 9: the statement of regulation of miRISC by IGF2BP1, FMR1 and ATXN2L lacks references, which should be added.

We have updated the references. Please note that IGF2BP1 did not pass the significance threshold of the new analysis and was replaced by RC3H2 (roquin-2).

8. The CDC25C interactome is rather descriptive and mainly correlated to similar functions of candidates or published literature. Since the selectivity of the method is analyzed here, it would be important to at least validate these interactors by additional independent approaches. Otherwise it is difficult to estimate whether the presented interactors are real or not. This needs to be

added.

As mentioned above, the addition of more replicates and applying the new analysis pipeline led to a somewhat different dataset. Seven proteins are now deemed to be in close proximity to the Cdc25C/14-3-3 ϵ complex. We have tried to validate two hits (CKAP5 and LMO7) for which we identified suitable antibodies. CKAP5 was not conclusive (absence of signal may also reflect a transient interaction) but LMO7 was found to associate with Cdc25c and 14-3-3 in co-IP experiments (new Fig. 2e, described on page 171-186). In the course of the revision of this paper, Xu X, et al (Nature Commun 2017) showed that LMO7, phosphorylated-Cdc25C and 14-3-3 ϵ form a trimeric complex that sequester Cdc25C in the cytoplasm. Hence one previously unknown interaction could be revealed by split-BioID. Since we are not a cell cycle focused lab and the main reason we chose to study the Cdc25C/14-3-3 pair was because it had been used before to validate another PCA, we did not investigate this further.

9. The validation of GIGYF2 is rather weak. The authors show weak interaction in co-IPs and a potential role in gene silencing is documented only by using one specific luciferase reporter. The effects are also not very strong. I understand that the main aim of the study is the development of Split-BioID but if the authors claim that they have identified a novel gene-silencing factor, they need to characterize it sufficiently. Localization studies could be added to show that the proteins are at least in the same compartments. Several positive controls should be added to Figure 4B to view the observed effects in the context of known gene silencing factors.

Thank you for your comments. We agree that the co-IP experiments show at best a weak interaction. We have addressed a similar concern from reviewer #2 (see response above) by adding an in vitro binding assay that demonstrate a direct interaction between the GYF domain of GIGYF2 and the PPGL motif within the C-terminal effector domain of TNRC6C (new Fig. 5c and 5d, described on page 8, lines 350-365). Of note, with this assay, GIGYF2 is also identified as the missing putative interacting protein to the conserved P-GL motif in zebrafish TNRC6A described in Mishima Y et al, *PNAS* 2012 and discussed in Braun J et al, *Cold Spring Harbor perspectives in biology* 2012. We have performed IF studies and show that Ago2 and GIGYF2 both show cytoplasmic localization (new Fig. 5a). We have added knock-down of the TNRC6 proteins and critical CNOT subunits as positive controls of the luciferase assay (new Fig. 6).

10. The authors state that GIGYF2 is part of a translation repression complex. I understand that 4EHP is not found in the Split-BioID data? The authors should nevertheless test in co-IPs whether this complex is involved in miRNA-guided translational repression. These factors should also be tested in the luciferase experiments shown in Figure 4B.

Indeed 4EHP was neither found in the split-BioID data nor in the Ago2-BioID data. We have identified an antibody suitable for detection of endogenous 4EHP, however the signals were weak and the antibody not suitable for IP.

The lab of N. Sonenberg showed that depleting 4EHP results in the co-depletion of GIGYF2 which complicates the interpretation of the luciferase assay. We nevertheless performed this experiment to challenge our data since depletion of 4EHP should indeed lead to a similar alleviation of miRNA-mediated repression to the one observed upon knockdown of GIGYF2 at least due to the co-depletion mentioned above. We did observe these similar effects on miRNA-mediated silencing upon 4EHP knockdown. However at this stage it is not possible to assign the effect to a direct effect of 4EHP or an indirect effect of the co-depletion of GIGYF2 (and possibly of GIGYF1). We present the data in this answer letter, however for the sake of space and clarity we would rather keep them out of the published manuscript as they do not add any useful information at this stage (we would need to perform many more experiments beyond the scope of our manuscript to make sure the effect is direct or not). Hence if reviewer 4 agrees, we would like to keep Fig. 6 as it is in the re-submitted manuscript (without the 4EHP knockdown data) and not as presented in this answer letter.

11. Figure 3D: BirA is split between Ago2 and TNRC6C, i.e. only proteins that are in proximity of an Ago2-TNRC6C complex will be biotinylated. Mechanistically, it is difficult to understand why TNRC6A and B are among the identified targets. The authors should at least present a model in which such interactions would make sense.

We have addressed this issue in the manuscript as follow (pages 6-7, lanes 275-279): most mRNA targets of miRNA have multiple miRNA binding sites and hence can be bond by multiple miRISC complexes if the TNRC6 proteins belonging to these miRISC complexes come in sufficient proximity within the cellular context they will be biotinylated. Similarly, all Ago (1-4) proteins are found in IPs of individual Ago proteins.

Another potential explanation might be that multimers of TNRC6 proteins exist within an individual miRISC complex. To our knowledge the stoichiometry of individual proteins with the miRISC is currently unknown and a miRISC has not yet been reconstituted in vitro. Hence this possibility cannot be excluded.

Reviewers' Comments:

Reviewer #1 (Remarks to the Author):

The manuscript 'Split-BioID: a conditional proteomics approach to monitor the composition of spatiotemporally defined protein complexes' by Schopp et. al., describes the creation and proof of principle application of binary protein complementation to the BioID system of proximity protein biotinylation. Overall the premise of split-BioID is sound and the major objective of creating a functionally reconstitutable set of BioID fragments was tested. The authors have made considerable efforts to address reviewers' critiques and considerably improved the manuscript, at least from the perspective of testing splitBioID, which should assist investigators seeking to utilize a similar technical approach. Additionally they have identified a novel protein that contributes to miRNA-mediated silencing.

Comments: Transient transfection is generally not recommended or utilized for BioID pull-down studies. While certainly useful for initial tests of fusion protein efficacy/targeting the highly variable and typically gross overexpression in those cells that are transfected leads to inappropriate behavior and/or localization of those proteins which confounds biological relevance. While I do not suggest that all of the transient transfection BioID pull-down studies in this manuscript need to be done using stably expressing cells, preferably those expressing a physiologically relevant level of the protein, I would like to see some discussion of the caveats with this approach and in the interpretation of BioID data obtained from transient transfection. The authors already discuss overexpression in lines 422-433 and perhaps this would be an ideal place to add caveats about transient transfection more explicitly to help guide future users of this splitBioID approach.

The MS results provided only seem to show the proteins that passed the described exclusion criteria. It would be useful for future users of this approach to be able to see the identities and levels of the background proteins that were excluded, perhaps in a separate tab within each worksheet.

Reviewer #2 (Remarks to the Author):

The authors adequately addressed my comments. The manuscript improved substantially because of the comments by the different reviewers. I still have a few minor comments:

Are all proteomics experiments analyzed in the same way? If so, can you also show the volcano plots for the 14-3-3ε/Cdc25C studies?

I also believe that the volcano plot(s) merit some space in the main figures as they are a good way to get a view on the proteomics data that is obtained with this approach.

I was referring to the following reference from the Ting group: Martell et al., *Nature Biotechnology* 34, 774–780 (2016). A split horseradish peroxidase for the detection of intercellular protein–protein interactions and sensitive visualization of synapses. I believe that this should be mentioned in the manuscript.

Reviewer #4 (Remarks to the Author):

Schopp et al. have addressed all points that I had raised on their previous version. They have adequately responded and new data added where necessary and possible. I am satisfied with the revised version of the manuscript.

REVIEWERS' COMMENTS:

Reviewer #1 (Remarks to the Author):

The manuscript 'Split-BioID: a conditional proteomics approach to monitor the composition of spatiotemporally defined protein complexes' by Schopp et. al., describes the creation and proof of principle application of binary protein complementation to the BioID system of proximity protein biotinylation. Overall the premise of split-BioID is sound and the major objective of creating a functionally reconstitutable set of BioID fragments was tested. The authors have made considerable efforts to address reviewers' critiques and considerably improved the manuscript, at least from the perspective of testing splitBioID, which should assist investigators seeking to utilize a similar technical approach. Additionally they have identified a novel protein that contributes to miRNA-mediated silencing.

Comments: Transient transfection is generally not recommended or utilized for BioID pull-down studies. While certainly useful for initial tests of fusion protein efficacy/targeting the highly variable and typically gross overexpression in those cells that are transfected leads to inappropriate behavior and/or localization of those proteins which confounds biological relevance. While I do not suggest that all of the transient transfection BioID pull-down studies in this manuscript need to be done using stably expressing cells, preferably those expressing a physiologically relevant level of the protein, I would like to see some discussion of the caveats with this approach and in the interpretation of BioID data obtained from transient transfection. The authors already discuss overexpression in lines 422-433 and perhaps this would be an ideal place to add caveats about transient transfection more explicitly to help guide future users of this splitBioID approach.

We have added the additional discussion on the potential pitfall of transient tranfection and advantages of stable transfection or genome editing strategy in the paragraph suggested by the reviewer.

The MS results provided only seem to show the proteins that passed the described exclusion criteria. It would be useful for future users of this approach to be able to see the identities and levels of the background proteins that were excluded, perhaps in a separate tab within each worksheet.

We have added the complete datasets as requested

Reviewer #2 (Remarks to the Author):

The authors adequately addressed my comments. The manuscript improved substantially because of the comments by the different reviewers. I still have a few minor comments:

Are all proteomics experiments analyzed in the same way? If so, can you also show the volcano plots for the 14-3-3 ϵ /Cdc25C studies?

I also believe that the volcano plot(s) merit some space in the main figures as

they are a good way to get a view on the proteomics data that is obtained with this approach.

All proteomics experiments were analyzed the same way. We have added or moved the volcano plots to the main figures as suggested.

I was referring to the following reference from the Ting group: Martell et al., Nature Biotechnology 34, 774–780 (2016). A split horseradish peroxidase for the detection of intercellular protein–protein interactions and sensitive visualization of synapses. I believe that this should be mentioned in the manuscript.

We were aware of that manuscript but while it is a very nice piece of work it is not related to proteomics studies. Indeed the Ting lab developed the engineered ascorbate peroxidases APEX and then APEX2 because HRP is not active in mammalian cytosol as discussed in these two papers from the Ting lab: Martell J et al, Nat Biotechnol. 2012 Nov; 30(11): 1143–1148 and Rhee H-W et al, Science. 2013 Mar 15; 339(6125): 1328–1331. In fact, even their proteomics study of extracellular synaptic clefts (Loh K et al, Cell. 2016 Aug 25;166(5):1295-1307) the Ting lab used APEX2 rather than HRP. Hence we believe that a split-APEX2 assay rather than a split-HRP assay would be comparable to the split-BioID approach. We thus kept the discussion on alternative techniques as in the last submitted version of the manuscript.